# Causal Abstractions of Neural Networks

**Atticus Geiger,** * **Hanson Lu,** * **Thomas Icard, and Christopher Potts**
Stanford
Stanford, CA 94305-2150
{atticusg, hansonlu, icard, cgpotts}@stanford.edu

## Abstract

Structural analysis methods (e.g., probing and feature attribution) are increasingly important tools for neural network analysis. We propose a new structural analysis method grounded in a formal theory of *causal abstraction* that provides rich characterizations of model-internal representations and their roles in input/output behavior. In this method, neural representations are aligned with variables in interpretable causal models, and then *interchange interventions* are used to experimentally verify that the neural representations have the causal properties of their aligned variables. We apply this method in a case study to analyze neural models trained on Multiply Quantified Natural Language Inference (MQNLI) corpus, a highly complex NLI dataset that was constructed with a tree-structured natural logic causal model. We discover that a BERT-based model with state-of-the-art performance successfully realizes parts of the natural logic model's causal structure, whereas a simpler baseline model fails to show any such structure, demonstrating that BERT representations encode the compositional structure of MQNLI.

## 1    Introduction

Explainability and interpretability have long been central issues for neural networks, and they have taken on renewed importance as such models are now ubiquitous in research and technology. Recent structural evaluation methods seek to reveal the internal structure of these "black box" models. Structural methods include probes, attributions (feature importance methods), and interventions (manipulations of model-internal states). These methods can complement standard behavioral techniques (e.g., performance on gold evaluation sets), and they can yield insights into how and why models make the predictions they do. However, these tools have their limitations, and it has often been assumed that more ambitious and systematic causal analysis of such models is beyond reach.

Although there is a sense in which neural networks are "black boxes", they have the virtue of being completely closed and controlled systems. This means that standard empirical challenges of causal inference due to lack of observability simply do not arise. The challenge is rather to identify high-level causal regularities that *abstract away* from irrelevant (but arbitrarily observable and manipulable) low-level details. Our contribution in this paper is to show that this challenge can be met. Drawing on recent innovations in the formal theory of causal abstraction [1, 2, 5, 22], we offer a methodology for meaningful causal explanations of neural network behavior.

Our methodology *causal abstraction analysis*[2] consists of three stages. (1) Formulate a hypothesis by defining a causal model that might explain network behavior. Candidate causal models can be naturally adapted from theoretical and empirical modeling work in linguistics and cognitive sciences. (2) Search for an alignment between neural representations in the network and variables in the

---

*equal contribution

[2]We provide tools for causal abstraction analysis at http://github.com/hansonhl/antra and the code base for this paper at http://github.com/atticusg/Interchange

high-level causal model. (3) Verify experimentally that the neural representations have the same causal properties as their aligned high-level variables using the *interchange intervention* method of Geiger et al. [11].

As a case study, we apply this methodology to LSTM-based and BERT-based natural language inference (NLI) models trained on the logically complex Multiply Quantified NLI (MQNLI) dataset of Geiger et al. [10]. This challenging dataset was constructed with a tree-structured natural logic causal model [17, 29, 14]. Our BERT-based model has the structure of a standard NLI classifier, and yet it is able to perform well on MQNLI (88%), a result Geiger et al. achieved only with highly customized task-specific models. By contrast, our LSTM-based model is much less successful (46%).

The obvious scientific question in this case study is what drives the success of the BERT-based model on this challenging task. To answer this we employ our methodology. (1) We formulate hypotheses by defining simplified variants of the natural logic causal model. (2) We search over potential alignments between neural representations in BERT and variables in our high-level causal models. (3) We perform interchange interventions on the BERT model for each alignment. We find that our BERT model partially realizes the causal structure of the natural logic causal model; crucially, the LSTM model does not. High-level causal explanation for system behavior is often considered a gold standard for interpretability, one that may be thought quixotic for complex neural models [16]. The point of our case study is to show that this high standard can be achieved.

We conclude by comparing our methodology to probing and the attribution method of integrated gradients [27]. We argue probing is unable to provide a causal characterization of models. We show formally that attribution methods do measure causal properties, and in that way they are similar to the tool of interchange interventions. However, our methodology of causal abstraction analysis provides a framework for systematically measuring and aggregating such causal properties in order to evaluate a precise hypothesis about abstract causal structure.

## 2 Related Work

**Probes**  Probes are generally supervised models trained on the internal representations of networks with the goal of determining what those internal representations encode [7, 13, 20, 28]. Probes are fundamentally unable to directly measure causal properties of neural representations, and Ravichander et al. [21], Elazar et al. [9], and Geiger et al. [11] have argued that probes are limited in their ability to provide even indirect evidence of causal properties.

We now present an analytic example in which probing identifies seemingly crucial information in representations that have no causal impact on behavior. We assume the structure of the simple addition network $N_+$ in Figure 1. For our embedding, we simply map every integer $i$ in $\mathbb{N}_9$ to the 1-dimensional vector $[i]$. The weight matrices are

$$W_1 = \begin{pmatrix} 1 \\ 1 \\ 0 \end{pmatrix} \quad W_2 = \begin{pmatrix} 1 \\ 1 \\ 1 \end{pmatrix} \quad W_3 = \begin{pmatrix} 0 \\ 0 \\ 1 \end{pmatrix} \quad \mathbf{w} = \begin{pmatrix} 0 \\ 1 \\ 0 \end{pmatrix}$$

The output for an input sequence $\mathbf{x} = (i, j, k)$ is given by $(\mathbf{x}W_1; \mathbf{x}W_2; \mathbf{x}W_3)\, \mathbf{w}$.

In this network, $\mathbf{x}W_1$ perfectly encodes $i + j$, and $\mathbf{x}W_3$ perfectly encodes $k$. Thus, the identity model probe will be perfect in probing those representations for this information. However, neither representation plays a causal role in the network behavior; only $\mathbf{x}W_2$ contributes to the output.

**Attribution Methods**  Attribution methods aim to quantify the degree to which a network representation contributes to the output prediction of the model, for a specific example or set of examples [3, 24, 26, 27, 32]. In contrast to probing, the well known integrated gradients method (IG) can be given an unambiguous causal interpretation. Following [27] we define the vector $IG(\mathbf{x})$, for an input $\mathbf{x}$ relative to a baseline $\mathbf{b}$, to have $i$th component $IG_i(\mathbf{x})$ given by the expression on the left:

$$(x_i - b_i) \cdot \int_{\alpha=0}^{1} \frac{\partial F(\alpha \mathbf{x} + (1-\alpha)\mathbf{b})}{\partial x_i} d\alpha \quad = \quad (x_i - b_i) \cdot \int_{\alpha=0}^{1} \lim_{\epsilon \to 0} \frac{F(\mathbf{x}^{\alpha,\epsilon}) - F(\mathbf{x}^{\alpha})}{\epsilon} d\alpha$$

Abbreviating the weighted average $\alpha \mathbf{x} + (1-\alpha)\mathbf{b}$ by $\mathbf{x}^{\alpha}$, letting $\mathbf{x}^{\alpha,\epsilon}$ be the vector that differs from $\mathbf{x}^{\alpha}$ in that the $i$th coordinate is increased by $\epsilon$, and then expanding the definition of partial derivative, this can be written in the form given on the right. The difference $F(\mathbf{x}^{\alpha,\epsilon}) - F(\mathbf{x}^{\alpha})$ is known in the

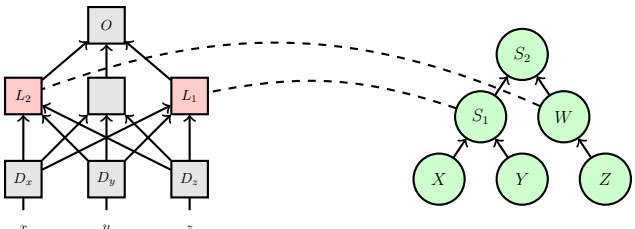

(a) The causal model $C_+$ (right) that first computes $S_1 = X + Y$ and $W = Z$, before computing the final output $S_2 = W + S_1$ aligned with the neural network $N_+$ (left) with $L_1$ highlighted as the hypothesized location encoding $S_1 = X + Y$ and $L_2$ as the location encoding $W = Z$.

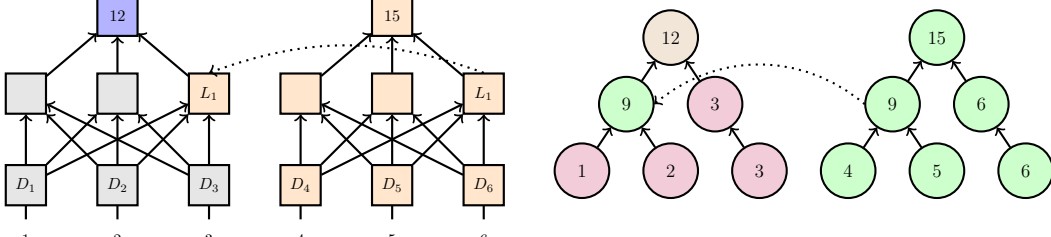

(b) Low-level neural network interchange intervention. The network processes two different input sequences. The neural representation created at $L_1$ for input sequence $(1, 2, 3)$ is replaced by the corresponding representation created for input sequence $(4, 5, 6)$.

(c) High-level symbolic computation interchange intervention. The computation processes two different input sequences. The sum at $S_1$ for input sequence $(1, 2, 3)$ is replaced by the corresponding representation created for input sequence $(4, 5, 6)$.

Figure 1: Our motivating example where we hypothesis that a symbolic computation $C_+$ is a causal abstraction of a neural network $N_+$ under a particular alignment (top). We can experimentally confirm this hypothesis by conducting an interchange intervention on both the network and the computation with every pair of inputs and evaluating whether the intervened network and intervened computation have the same counterfactual output behavior. We schematically depict an interchange intervention on the network $N_+$ (bottom left) and the computation $C_+$ (bottom right) with the base input $(1, 2, 3)$ and the source input $(4, 5, 6)$. Observe that the output of the intervened neural network matches the output of the intervened symbolic computation, so we have success for this pair of inputs.

causal literature as the (individual) *causal effect* on the output (e.g., [15]) of increasing neuron $i$ by $\epsilon$ relative to the fixed input $\mathbf{x}^\alpha$. So, essentially, $IG_i(\mathbf{x})$ is measuring the average "limiting" causal effect of increasing neuron $i$ along the straight line from the baseline vector to the input vector $\mathbf{x}$, weighted by the difference at $i$ between input and baseline. More recently, Chattopadhyay et al. [6] develop an attribution method that explicitly treats neural models as structured causal models and directly computes the individual causal effect of a feature to determine its attribution.

Attribution methods can measure causal properties, and, in that way, they are similar to the tool of interchange interventions. However, our methodology of causal abstraction analysis provides a framework for systematically measuring and aggregating such causal properties in order to evaluate a precise hypothesis about abstract causal structure.

**Causal Abstraction** Our goal is to evaluate whether the internal structure of a neural network realizes an abstract causal process. To concretize this, we turn to formal, broadly interventionist theories of causality [25, 19], in which causal processes are characterized by effects of interventions, and theories of abstraction [1, 2, 5, 22] where relationships between two causal processes are determined by the presence of systematic correspondences between the effects of interventions.

The notion of abstraction that we employ here is a relatively simple one called *constructive abstraction* [1]. Informally, a high-level model is a constructive abstraction of a low-level model if there is a way to partition the variables in the low-level model where each high-level variable can be assigned to a low-level partition cell, such that there is a systematic correspondence between interventions on the low-level partition cells and interventions on the high-level variables.

There are two properties of constructive abstraction that make it ideal for neural network analysis. First, the information content of partition cells of low-level variables can be determined by the high-level variables that they correspond to. For neural networks, the partition cells of low-level variables are sets of neurons, and our method supports reasoning at the level of vector representations (sets of neurons). Second, the causal dependencies between partitions of low-level variables are not necessarily preserved as causal dependencies between the high-level variables corresponding to these partitions. For example, the low-level model might be a fully connected neural network, whereas the high-level model might have much sparser connections. For neural network analysis, this means we can find causal abstractions that have far simpler causal structures than the underlying neural networks. We provide an example in the next section.

## 3 Causal Abstraction Analysis of Neural Networks

We now describe our methodology in more detail, illustrating the relevant concepts with an example of a neural network performing basic arithmetic. Specifically, suppose that we have a neural network $N_+$ that takes in three vector representations $D_x, D_y, D_z$ representing the integers $x$, $y$, and $z$, and outputs the sum of the three inputs: $N_+(D_x, D_y, D_z) = x + y + z$. We seek an informative causal explanation of this network's behavior.

**Formulating a Hypothesis** A human performing this task might follow an algorithm in which they add together the first two numbers and then add that sum to the third number. We can hypothesize that the behavior of $N_+$ is explained by this symbolic computation. Specifically, the network combines $D_x$ and $D_y$ to create an internal representation at some location $L_1$ encoding $x + y$; it encodes $z$ at some location $L_2$; and $L_1$ and $L_2$ are composed to encode $a + z$ at the location of the output representation. This hypothesis is given schematically in Figure 1a.

Following our methodology, we first define the causal model $C_+$ in Figure 1a. Our informal hypothesis that a neural network's behavior is explained by a simple algorithm can then be restated more formally: $C_+$ is a constructive abstraction of the neural network $N_+$.

**Alignment Search** Now that we have hypothesized that the causal model $C_+$ is a causal abstraction of the network $N_+$, the next step is to align the neural representations in $N_+$ with the variables in $C_+$. The input embeddings $D_x$, $D_y$, and $D_z$ must be aligned with the input variables $X$, $Y$, and $Z$ and the output neuron $O$ must be aligned with the output variable $S_2$. That leaves the intermediate variables $S_1$ and $W$ to be aligned with neural representations at some undetermined locations $L_1$ and $L_2$. If this were an actual experiment (see below), we would perform an *alignment search* to consider many possible values for $L_1$ and $L_2$. Each alignment is a hypothesis about where the network $N_+$ stores and uses the values of $S_1$ and $W$. For the example, we assume the alignment in Figure 1a.

**Interchange Interventions** Finally, for a given alignment, we experimentally determine whether the neural representations at $L_1$ and $L_2$ have the same causal properties as $S_1$ and $W$. The basic experimental technique is an *interchange intervention*, in which a neural representation created during prediction on a "base" input is interchanged with the representation created for a "source" input [11]. We now show informally that this method can be used to prove that the causal model $C_+$ is a constructive abstraction of the neural network $N_+$ (Appendix G has formal details).

We first intervene on the causal model. Consider two inputs $\mathbf{a}, \mathbf{a}' \in (\mathbb{N}_9)^3$ where $\mathbb{N}_9$ is the set of integers 0–9. Let $\mathbf{a} = (x, y, z)$ be the base input and $\mathbf{a}' = (x', y', z')$ be the source input. Define

$$C_+^{S_1 \leftarrow \mathbf{a}'}(\mathbf{a}) = x' + y' + z \tag{1}$$

to be the output provided by $C_+$ when $S_1$, the variable representing the intermediate sum, is intervened on and set to the value $x' + y'$. Thus, for example, if the base input is $C_+(1, 2, 3) = 6$, and the source input is $\mathbf{a}' = (4, 5, 6)$, then $C_+^{S_1 \leftarrow \mathbf{a}'}(1, 2, 3) = 4 + 5 + 3 = 12$. This process is depicted in Figure 1c.

Next, we intervene on the neural network $N_+$. Let $\mathbf{D}$ be an embedding space that provides unique representations for $\mathbb{N}_9$, and consider two inputs $D = (D_x, D_y, D_z)$ and $D' = (D_{x'}, D_{y'}, D_{z'})$, where all $D_i$ and $D_{i'}$ are drawn from $\mathbf{D}$. In parallel with (1), define

$$N_+^{L_1 \leftarrow D'}(D) \tag{2}$$

to be the output provided by $N_+$ processing the input $D$ when the representation at location $L_1$ is replaced with the representation at location $L_1$ created when $N_+$ is processing the input $D'$. This process is depicted in Figure 1b.

With these two definitions, we can define what it means to test the hypothesis that $N_+$ computes $x + y$ at position $L_1$. Where $D_{\mathbf{a}}$ is an embedding for $\mathbf{a}$ and $D_{\mathbf{a}'}$ is an embedding for $\mathbf{a}'$, we test:

$$C_+^{S_1 \leftarrow \mathbf{a}'}(\mathbf{a}) = N_+^{L_1 \leftarrow D_{\mathbf{a}'}}(D_{\mathbf{a}}) \tag{3}$$

If this equality holds for all source and base inputs $\mathbf{a}$ and $\mathbf{a}'$, then we can conclude that, for every intervention on $S_1$, there is an equivalent intervention on $L_1$. If we can establish a corresponding claim for $W$ and $L_2$, then we have shown that $C_+$ is a constructive abstraction of $N_+$, since the inputs' relationships are established by our embedding and there are no other interventions on $C_+$ to test.

**Analysis** Suppose that all of our intervention experiments verify our hypothesis that $C_+$ is a constructive abstraction of $N_+$ with variables $S_1$ and $W$ aligned to neural representations at $L_1$ and $L_2$. This explains network behavior by resolving two crucial questions.

First, we learn what information is encoded in the representations $L_1$ and $L_2$. Neural representations encode the values of the high-level variables they are aligned with. The location $L_1$ encodes the variable $S_1$ and the location $L_2$ encodes the variable $W$. This is similar to what probing achieves. However, our method is crucially different from probing. In probing, information content is established through purely correlational properties, meaning a neural representation with no causal role in network behavior can be successfully probed, as we showed in Section 2. In causal abstraction analysis, information content is established through purely causal properties, ensuring that the neural representation is actually implicated in model behavior.

Second, we learn what causal role $L_1$ and $L_2$ play in network behavior. Neural representations play a parallel causal role to their aligned high-level variables. At the location $L_1$, $D_x$ and $D_y$ are composed to form a neural representation with content $x + y$ that is then composed with $L_2$ to create an output. The fact that $S_1$ doesn't depend on $z$ tells us that while $L_1$ depends on $D_z$ and representations at $L_1$ may even correlate with $z$, the information about $z$ is not causally represented at $L_1$. At the location $L_2$, the value of $z$ is simply repeated and then composed with $L_1$ to create a final output.

Our method assigns causally impactful information content, but also identifies the abstract causal structure along which representations are composed. It thus encompasses and improves on both correlational (probing) and attribution methods.

## 4    The Natural Language Inference Task and Models

**Multiply Quantified NLI Dataset** The Multiply Quantified NLI (MQNLI) dataset of Geiger et al. [10] contains templatically generated English-language NLI examples that involve very complex interactions between quantifiers, negation, and modifiers. We provide a few examples in Figure 2b; the empty-string symbol $\varepsilon$ ensures perfect alignments at the token level both between premises and hypotheses and across all examples.

The MQNLI examples are labeled using an algorithmic implementation of the natural logic of MacCartney and Manning [18] over tree structures, and MQNLI has train/dev/test splits that vary in their difficulty. In the hardest setting, the train set is provably the minimal set of examples required to ensure that the dev and test sets can be perfectly solved by a simple symbolic model; in the easier settings, the train set redundantly encodes necessary information, which might allow a model to perform perfectly in assessment by memorization despite not having found a truly general solution. For a fuller review of the dataset, see Appendix A.

MQNLI is a fitting benchmark given our goals for a few reasons. First, we can focus on the hardest splits that can be generated, which will stress-test our NLI architectures in a standard behavioral way. Second, the MQNLI labeling algorithm itself suggests an appropriate causal model of the data-generating process. Figure 2a summarizes this model in tree form, and it is presented in full detail in Geiger et al. [10]. This allows us to rigorously assess whether a neural network has learned to implement variants of this causal model. The complexity of the MQNLI examples creates many opportunities to do this in linguistically interesting ways.

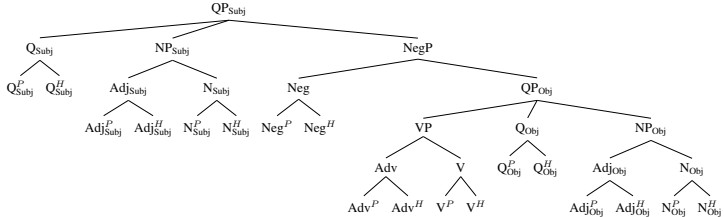

(a) The causal structure of the high-level natural logic causal model $C_{NatLog}$ that performs inference on MQNLI. The superscripts $P$ and $H$ stand for 'premise' and 'hypothesis' and the subscripts 'Subj' and 'Obj' stand for 'Subject' and 'Object'. The node labels are used to explain the experimental results in Section 5

$\varepsilon$ every $\varepsilon$ baker $\varepsilon$ $\varepsilon$ $\varepsilon$ eats $\varepsilon$ no $\varepsilon$ bread
**contradiction**
$\varepsilon$ no angry baker $\varepsilon$ $\varepsilon$ $\varepsilon$ eats $\varepsilon$ no $\varepsilon$ bread

$\varepsilon$ every silly professor $\varepsilon$ $\varepsilon$ $\varepsilon$ sells not every $\varepsilon$ book
**neutral**
$\varepsilon$ every silly professor $\varepsilon$ $\varepsilon$ $\varepsilon$ sells not every $\varepsilon$ chair

not every sad baker $\varepsilon$ $\varepsilon$ fairly admits not every odd idea
**entailment**
$\varepsilon$ some $\varepsilon$ baker does not $\varepsilon$ admits $\varepsilon$ no $\varepsilon$ idea

(b) MQNLI examples. The $\varepsilon$ token serves as padding (but still attended to by the model) and ensures a perfect alignment between both premises and hypotheses and across all examples. It is semantically an identity element.

| Model | Train | Dev | Test |
|---|---|---|---|
| CBoW | 88.04 | 54.18 | 53.99 |
| TreeNN | 67.01 | 54.01 | 53.73 |
| CompTreeNN | 99.65 | 80.17 | 80.21 |
| BiLSTM | 99.42 | 46.41 | 46.32 |
| BERT | 99.99 | **88.25** | **88.50** |

(c) MQNLI results. The first three models are from Geiger et al. 10, where the CompTreeNN is a task-specific model not suitable for general NLI and functions as an idealized upperbound. Our results show that BERT-based models can surpass this without such alignments.

Figure 2: The natural logic causal model (top), MQNLI examples (left) and MQNLI results (right).

**Models** We evaluated two models on MQNLI: a randomly initialized multilayered Bidirectional LSTM (BiLSTM; [23]) and a BERT-based classifier model in which the English `bert-base` parameters [8] are fine-tuned on the MQNLI train set. Output predictions are computed using the final representation above the [CLS] token. Models are trained to predict the relation of every pair of aligned phrases in Figure 2a. Additional model and training details are given in Appendix B.

**Results** Figure 2c summarizes the results of our BERT and BiLSTM models on the hardest fair generalization task Geiger et al. [10] creates with MQNLI. We find that our BiLSTM model is not able to learn this task, and that our BERT model is able to achieve high accuracy. The only models in Geiger et al. [10] able to achieve above 50% accuracy were task-specific tree-structured models with the structure of the tree in Figure 2a. Thus, our BERT-based model is the first general-purpose model able to achieve good performance on this hard generalization task. Without pretraining, the BERT-based model achieves ≈49.1%, confirming that pretraining is essential, as expected.

A natural hypothesis is that the BERT-based model achieves this high performance *because* it has in effect induced some approximation to the tree-like structure of the data-generating process in its own internal layers. With causal abstraction analysis, we are actually in a position to test this hypothesis.

## 5 A Case Study in Structural Neural Network Analysis

### 5.1 Causal Abstractions of Neural NLI models

**Formulating Our Hypotheses** We proceed just as we did for the simple motivating example in Section 3, except that we are now seeking to assess the extent to which the natural logic algebra in Figure 2a is a causal abstraction of the trained neural models in the above section.

The hallmark of Figure 2a is that it defines an alignment between premise and hypothesis at both lexical and phrasal levels. This permits us to run interchange interventions in a naturally compositional way. For a given non-leaf node $N$ in Figure 2a, let $C_{NatLog}^N$ be a submodel of $C_{NatLog}$ that computes the relation between the aligned phrases under $N$ and uses them to compute the final output relation between premise and hypothesis. For example, let $C_{NatLog}^{NP_{Obj}}$ be the submodel of $C_{NatLog}$ that computes

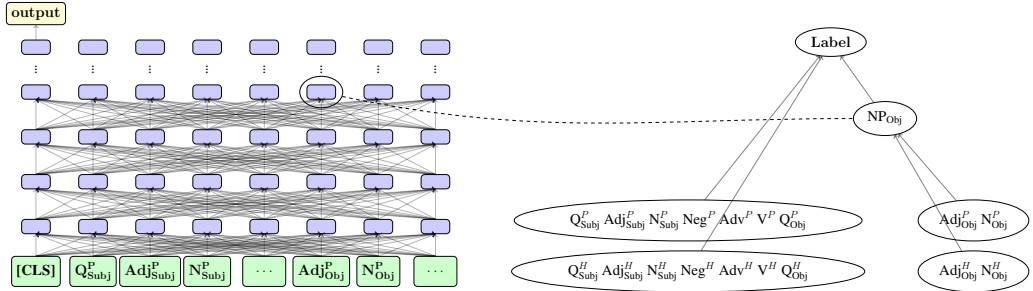

Figure 3: A BERT-based NLI model (left) aligned with the natural logic causal model $C_{NatLog}^{NP_{Obj}}$ (right), where the fourth vector representation above the $Adj_{Obj}^P$ token in the network is aligned with $NP_{Obj}$, the variable representing the relation between the object noun phrases. When analyzing a sample of 1000 examples, we found a subset of 383 where $C_{NatLog}^{NP_{Obj}}$ is an abstraction of $N_{NLI}$ under this alignment.

the relation between the two aligned object noun phrases and then uses that relation in computing the final output relation between premise and hypothesis (see Figure 3 right). We would like to ask whether our trained neural models also compute this relation between object noun phrases and use it to make a final prediction. We can pose this same question for other nodes which correspond to a pair of aligned subphrases.

**Alignment Search** For each $N$, we search for an alignment between a neural representation in $N_{NLI}$ and the variable $N$ in $C_{NatLog}^N$. In principle, any location in the network could be the right one for any causal model. Testing every hypothesis in this space would be intractable. Thus, for each $C_{NatLog}^N$, we consider a restricted set of hidden representations based on the identity of $N$. The BERT model we use has 12 Transformer layers [30], meaning that there are 12 hidden representations for each input token. Each alignment search considers aligning the intermediate high-level variable with dozens of possible locations in the grid of BERT representations. Specifically, the following locations were considered for each $N$:

- $Q_{Subj}$, $Adj_{Subj}$, $N_{Subj}$, Neg, Adv, V, $Q_{Obj}$, $Adj_{Obj}$, $N_{Obj}$: hidden representations above the two descendant leaf tokens.

- $NP_{Subj}$, VP, and $NP_{Obj}$: same but above the four descendant leaf tokens.

- $QP_{Obj}$: hidden representations above $Q_{Obj}^P$ and $Q_{Obj}^H$.

- NegP: same but above $Neg^P$ and $Neg^H$.

- All nodes (for BERT): same but above [CLS] and [SEP].

For each alignment considered, we performed a full causal abstraction analysis. We report the results from the best alignments in Table 1, and we summarize the results from all alignments in Appendix D.

**Interchange Interventions** We first focus on our high-level causal models. Consider a non-leaf node $N$ from Figure 2a and two input token sequences $e$ and $e'$ from MQNLI. Define

$$C_{NatLog}^{N \leftarrow e'}(e) \qquad (4)$$

to be the output provided by the causal model $C_{NatLog}^N$ when processing input $e$ where the relation between the aligned subphrases under the node $N$ is changed to the relation between those subphrases in $e'$. For example, simplifying for the sake of exposition, suppose $e$ is (*some happy baker*, *no $\epsilon$ baker*), which has output label **contradiction**, and suppose $e'$ is (*every happy person*, *some happy baker*), which has output label **entailment**. We wish to intervene on the noun phrase, so $N = $ NP. In $e$, the noun phrase relation is entailment; in $e'$, it is reverse entailment. Thus, $C_{NatLog}^{NP \leftarrow e'}(e)$ changes the object noun phrase relation in $e$ to entailment while holding everything else about $e$ constant. This results in the output label for the example (*some happy person*, *no $\epsilon$ baker*), which is **neutral**.

Next, we consider interventions in a neural model $N_{NLI}$. Define

$$N_{NLI}^{L \leftarrow e'}(e) \qquad (5)$$

Table 1: Largest subsets of examples on which specific models $C_{NatLog}^N$ are abstractions of an LSTM and BERT model trained on MQNLI. We record the size of such subsets as a percentage of the total 1000 examples. On this subset, we know that the neural models compute a representation of the relation between the aligned subphrases under $N$ and use this information to make a final prediction.

| Causal Model | LSTM | BERT |
|---|---|---|
| $Q_{Subj}$ | 0.7 | 13.1 |
| $Q_{Obj}$ | 0.9 | 7.3 |
| Neg | 0.7 | 21.4 |
| $Adj_{Subj}$ | 2.5 | 6.7 |
| $N_{Subj}$ | 1.2 | 5.5 |
| $Adj_{Obj}$ | 0.9 | 14.1 |
| $N_{Obj}$ | 0.7 | 8.8 |
| V | 0.4 | 11.4 |
| Adv | 1.4 | 7.9 |
| $NP_{Subj}$ | 1.0 | 6.7 |
| $NP_{Obj}$ | 0.7 | **38.3** |
| VP | 0.4 | 11.4 |
| NegP | 0.9 | 11.8 |

| Nodes removed | BERT |
|---|---|
| $N_{Obj}^H$ | 31.9 |
| $A_{Obj}^H$ | 15.7 |
| $N_{Obj}^P$ | 33.8 |
| $A_{Obj}^P$ | 15.8 |
| $N_{Obj}^H, A_{Obj}^H$ | 31.9 |
| $N_{Obj}^H, N_{Obj}^P$ | 14.1 |
| $N_{Obj}^H, A_{Obj}^P$ | 32.2 |
| $N_{Obj}^P, A_{Obj}^H$ | 31.6 |
| $A_{Obj}^H, A_{Obj}^P$ | 8.8 |
| $N_{Obj}^P, A_{Obj}^P$ | 32.1 |

| Nodes added | BERT |
|---|---|
| $Adj_{Subj}^P$ | 30.5 |
| $N_{Subj}^P$ | 37.2 |
| $Neg^P$ | 14.9 |
| $Adv^P$ | 26.9 |
| $V^P$ | 35.6 |
| $Q_{Obj}^H$ | 16.2 |
| $Adj_{Subj}^H$ | 13.4 |
| $N_{Subj}^H$ | 12.0 |
| $Neg^H$ | 34.4 |
| $Adv^H$ | 16.2 |
| $V^H$ | 13.4 |
| $Q_{Obj}^H$ | 12.0 |

(a) Main results (clique sizes) for non-leaf nodes of the tree in Figure 2a. The hypothesis we have most evidence for is that the BERT model computes a representation of the $NP_{Obj}$ node with the alignment shown in Figure 3. Remarkably, with 1000 examples sampled, we found a subset of 383 examples where $C_{NatLog}^{NP_{Obj}}$ is an abstraction of BERT.

(b) Detailed results (clique sizes) for Alternative causal models in a "neighborhood" around the model $C_{NatLog}^{NP_{Obj}}$, which has a single intermediate variable composed of four lexical items (See Figure 3). At left, we have alternative causal models where one or two of those lexical items are removed from the composition. At right, we have alternatives obtained by adding one lexical item to the composition. We observe that no alternative hypothesis about causal structure considered has more evidence.

to be the output provided by $N_{NLI}$ processing the input $e$ when the representation at location $L$ is replaced with the representation at location $L$ created when $N_{NLI}$ is processing $e'$. This is exactly the process depicted in Figure 1, except now the networks are the complex trained networks of Section 4.

Our hypothesis linking Figure 2a with a model $N_{NLI}$ takes the same form as (3). The causal model $C_{NatLog}^N$ is a constructive abstraction of $N_{NLI}$ when, for some representation location $L$, it is the case that, for all MQNLI examples $e$ and $e'$, we have

$$C_{NatLog}^{N \leftarrow e'}(e) = N_{NLI}^{L \leftarrow e'}(e) \tag{6}$$

This asserts a correspondence between interventions on the representations at $L$ in network $N_{NLI}$ and interventions on the variable $N$ in the causal model $C_{NatLog}^N$. If it holds, then $N_{NLI}$ computes the relation between the aligned phrases under the node $N$ and uses this information to compute the relation between the premise and hypothesis.

We call a pair of examples $(e, e')$ *successful* if it satisfies equation (6), i.e., interventions in both the target causal model and neural model produce equal results. In addition, to isolate the causal impact of our interventions, we specifically focus on pairs $(e, e')$ for which performing the intervention produces a different output value than without the intervention. We call a pair $(e, e')$ *impactful* if:

$$C_{NatLog}^{N \leftarrow e'}(e) \neq C_{NatLog}^N(e) \tag{7}$$

**Quantifying Partial Success** Equation (6) universally quantifies over all examples. We do not expect this kind of perfect correspondence to emerge in practice for real problems: neural network training is often approximate and variable in nature, and even our best model does not achieve *perfect* performance. However, we can still ask how widely (6) holds for a given model. To do this, we seek to find the *largest subset* of MQNLI on which $C_{NatLog}^N$ is an abstraction of our neural models, for each non-leaf node $N$ in $C_{NatLog}$.

More specifically, considering each example in MQNLI as a vertex in a graph, we add an undirected edge between two examples $e_i$ and $e_j$ if and only if both the ordered pairs $(e_i, e_j)$ and $(e_j, e_i)$ satisfy (6). In other words, $C_{NatLog}^N$ is an abstraction of a neural model on a subset of examples $S$ of MQNLI if and only if all examples in $S$ form a *clique*.

The number of interventions we need to run scales quadratically with the number of inputs we consider, so we sample 1000 MQNLI examples, producing a total of $1000^2 = 1M$ ordered pairs. We only consider examples for which the neural network outputs a correct label. For each node $N$ and each of its corresponding neural network locations $L$, we perform interventions on all of these pairs.

We choose to measure the largest clique with at least one impactful edge, because (1) the causal abstraction relation holds with full force on that clique, but other measures such as the total number of connections lack this theoretical grounding, and (2) if a clique has at least one impactful edge, that guarantees the high-level variable is being used.

**Results and Analysis**    For each target causal model node $N$ and neural network representation location $L$, we construct a graph as described above with 1000 examples as vertices and add an edge between two examples $e_i$ and $e_j$ if and only if *both* $(e_i, e_j)$ and $(e_j, e_i)$ are successful. We then find the largest clique in this graph with at least one impactful edge and record its size.

Table 1a shows, for each causal model node $N$, the maximum size of cliques found among all neural locations. With this stricter *impactful* criterion (as opposed to simply using intervention success), our results show that, for almost all nodes $N$, our target causal model $C_{NatLog}^N$ is indeed a causal abstraction of BERT on a significant number of examples in our dataset. These subsets are much smaller for the BiLSTM model.

We also investigated alternative high-level causal structures that are not variants of $C_{NatLog}$ from Figure 2a. Specifically, we consider alternative models in a "neighborhood" around the model $C_{NatLog}^{\mathrm{NP_{Obj}}}$ that can be obtained by adding one leaf, or by removing one or two leaves to the composition. These results are in Table 1b. Remarkably, all of these alternative models result in smaller clique sizes, significantly so for many of them. This further supports the significance of our results.

This analysis is similar to the analysis of our hypothetical addition example in Section 3, except for two crucial differences. First, for each variable $N$, we are hypothesizing that the causal model $C_{NatLog}^N$ is an abstraction of $N_{NLI}$, whereas in the addition example there was only one model. To investigate this difference, we take $N = \mathrm{NP_{Obj}}$ as a paradigm case, as it is the model with the strongest results. (The results for other nodes are in Appendix D.) Second, we only achieved partial experimental success, whereas in the addition example we assumed complete success. Crucially, this means that the following analysis will be valid only on subsets of the input space on which the abstraction relation holds between $N_{NLI}$ and $C_{NatLog}^{\mathrm{NP_{Obj}}}$.

We visualize the results of our intervention experiments for the node $\mathrm{NP_{Obj}}$ in Figure 4. The alignment with the largest subset of inputs aligns the $\mathrm{NP_{Obj}}$ variable in $C_{NatLog}^{\mathrm{NP_{Obj}}}$ with the neural representation on the fourth layer of BERT above the $\mathrm{Adj}_{\mathrm{Obj}}^P$ token (see Figure 3). Because neural representations encode the value of their aligned variables and play a parallel causal role to their high-level variables, we know that, on this subset of input examples, at the fourth neural representation above the $\mathrm{Adj}_{\mathrm{Obj}}^P$ token, the four input embeddings for the object nouns and adjectives in the premise and hypothesis are composed to form a neural representation with information content of the relation between the object noun phrases in the premise and hypothesis. Then this representation is composed with the other input-embeddings to create an output representing the relation between the premise and hypothesis.

## 5.2    Comparison with Other Structural Analysis Methods

**Probes**    We probed neural representation locations for the relation between aligned subexpressions on a subset of 12,800 randomly selected MQNLI examples. For a pair of aligned subexpressions below a node $N$ in Figure 2a, we probe the columns above the same set of restricted class of tokens as described in Section 5.1.

To evaluate these probes, we report accuracy as well as *selectivity* as defined by Hewitt and Liang [12]: probe accuracy minus control accuracy, where *control accuracy* is the train set accuracy of a probe with the same architecture but trained on a control task to factor out probe success that can be

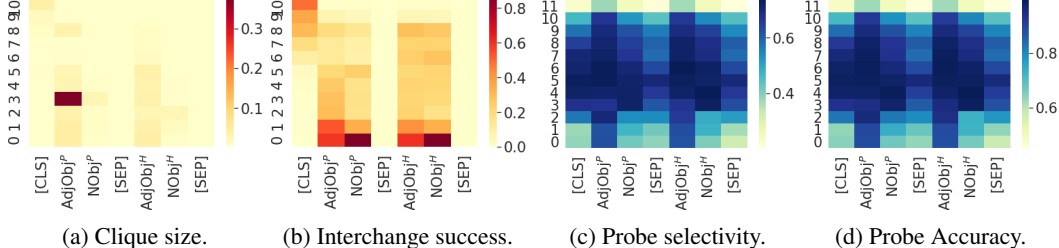

| (a) Clique size. | (b) Interchange success. | (c) Probe selectivity. | (d) Probe Accuracy. |

Figure 4: Interchange intervention and probing results for the $NP_{Obj}$ position. Vertical axes denote layers of BERT and horizontal axes denote the token position of hidden representations. The intervention success rates reported here are calculated based on intervention experiments with a change in the output label. Clique sizes are reported as % of 1000 examples.

attributed to the probe model itself. Our control task is to learn a random mapping from node types to semantic relations; see Appendix C for full details on how this task was constructed.

Figure 4 summarizes our probing results for $N = NP_{Obj}$, along with corresponding interchange intervention results for comparison. Probes tell us that information about the relation between the aligned noun phrases is encoded in nearly all of the locations we considered, and using the selectivity metric does not result in any qualitative change. In contrast, our intervention heatmaps indicate only a small number of locations store this information in a causally relevant way. Clearly, our intervention experiments are far more discriminating than probes. Appendix D provides examples involving other variables along with the intervention experiments, where the general trend of interchange interventions being more discriminating holds.

**Integrated Gradients** Attribution methods that estimate feature importance can measure causal properties of neural representations, but a single feature importance method is an impoverished characterization of a representation's role in network behavior. Whereas our interchange interventions gave us high-level information about how a neural representation is composed and what it is composed into, attribution methods simply tell us "how much" a representation contributes to the network output on a give input. Moreover, intervention interchanges provide a rich, high-level characterization of causal structure on a space of inputs.

We use integrated gradients on our models to verify the intuitive hypothesis that if a premise and hypothesis differ by a single token, then the neural representations above that token should be more causally responsible for the network output than other representations. For example, given premise 'Every sleepy cat meows' and hypothesis 'Some hungry cat meows', the attributive modifier position is different and the rest are matched. The neural representations above the adjective tokens *sleepy* and *hungry* should be more important for the network output than others, because if those adjectives were the same, the example label would change from **neutral** to **entailment**. We summarize the results of our integrated gradient experiments in Appendix E, where we confirm our intuitive hypothesis.

## 6    Conclusion

We have introduced a methodology for deriving interpretable causal explanations of neural network behaviors, grounded in a formal theory of causal abstraction. The methodology involves first *formulating a hypothesis* in the form of a high-level, interpretable causal model, then *searching for an alignment* between the neural network and the causal model, and finally *verifying experimentally* that the neural representations encode the same causal properties and information content as the corresponding components of the high-level causal model. As a case study demonstrating the feasibility of the approach, we analyzed neural models trained on the semantically formidable MQNLI dataset. Guided by the intuition that success on this challenging task may call for a way of recapitulating the causal structure of the natural logic model that generates the MQNLI data, we were able to verify the hypothesis that a state-of-the-art BERT-based model partially realizes this structure, whereas baseline models that do not perform as well fail to do so. This suggestive case study demonstrates that our theoretically grounded methodology can work in practice.

## Acknowledgments and Disclosure of Funding

Our thanks to Amir Feder, Noah Goodman, Elisa Kreiss, Josh Rozner, Zhengxuan Wu, and our anonymous reviewers. This research is supported in part by grants from Facebook and Google.

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
