# A    Additional Details on MQNLI

## A.1    Dataset Description

The MQNLI dataset contains sentences of the form

$$Q_S \; Adj_S \; N_S \; Neg \; Adv \; V \; Q_O \; Adj_O \; N_O$$

where $N_S$ and $N_O$ are nouns, V is a verb, $Adj_S$ and $Adj_O$ are adjectives, and Adv is an adverb. These categories all have 100 words. Neg is *does not*, and $Q_S$ and $Q_O$ can be *every*, *not every*, *some*, or *no*. Additionally, $Adj_S$, $Adj_O$, Adv, and Neg can be the empty string $\varepsilon$.

NLI examples are constructed so that non-identical non-empty nouns, adjectives, verbs, and adverbs with identical positions in $s_p$ and $s_h$ are semantically unrelated. This means that the learning task is trivial for these lexical items, as the correct relation is equivalence when they are identical and independence when they are not identical.

For our experiments, we used a train set with 500K examples, a dev set with 60k examples, and a test set with 10K examples – the most difficult generalization scheme of Geiger et al. [10].

## A.2    A Natural Logic Causal Model

Geiger et al. [10] construct a natural logic model that solves MQNLI using a formalization they call *composition trees*, which is easily translated into the causal model we call $C_{NatLog}$. Natural logic is a flexible approach to doing logical inference directly on natural language expressions [14, 17, 29] where the *semantic relations* between phrases are compositionally computed from the semantic relations between aligned subphrases and *projectivity signatures*, which encode how semantic operators interact compositionally with their arguments (which are semantic relations). The causal model $C_{NatLog}$ performs inference on aligned semantic parse trees that represent both the premise and hypothesis as a single structure and calculates semantic relations between all subphrases compositionally.

# B    Model Training and Interchange Experiment Details

We evaluated two models on MQNLI: a multi-layered bidirectional LSTM baseline and a Transformer-based model trained to do masked language modeling and next-sentence prediction [8]. We rely on the uncased BERT-base initial parameters from Hugging Face `transformers` [31]. For both models, we concatenate the premise $s_p$ and hypothesis $s_h$ into one string with special separator tokens: [CLS] $s_p$ [SEP] $s_h$ [SEP].

For the BiLSTM, we concatenate the hidden state above the last [SEP] and the [CLS] in the last layer for the forward and backward directions respectively to obtain a representation for the whole input, and then apply three linear transformations on top of that. The final transformation outputs a logit score for each class in the label space.

For the BERT model, we apply one linear transformation to the final layer's hidden representation above the [CLS] token to obtain a logit score for each label class.

## B.1    Tokenization

In the original setting of MQNLI, some positions in the premise and hypothesis consist of two words such as *not every* in $Q_S$ and $Q_O$ and *does not* in the leaf nodes $Neg^P$ and $Neg^H$ (as shown in the beginning of Section A.1). We treat them as two separate tokens in order to utilize BERT's knowledge of these function words. To ensure all sentences have identical length, we introduce one extra empty string tokens $\varepsilon$ to single-word quantifiers and two such tokens in the place of $Neg^P$ and $Neg^H$ for sentences without negation.

For consistency, we use the same tokenization method for both models.

Table 2: Ablation results.

| Model | Dev | Test |
|---|---|---|
| Fine-tuned BERT | **88.25** | **88.50** |
| Without augmented examples | 55.42 | 54.51 |

## B.2 Dataset Augmentation with Labeled Subphrases

The *hard but fair* MQNLI generalization task requires the dataset to explicitly expose the model to labels for each intermediate node that is a relation in $C_{NatLog}$. For each training example $(s_p, s_h, y) \in \mathcal{S}$, we create an additional example $(s_p^N, s_h^N, y^N)$ for each node $N$. $(s_p^N, s_h^N)$ is a *subphrase* pair made up of all the leaf tokens under node $N$ in the original input $(s_p, s_h)$, and $y^N$ is the relation computed by $C_{NatLog}$ for that subphrase pair. The set of labels we use for these subphrase examples is disjoint from that of the full-sentence examples. During training, the augmented examples are coupled with original examples in each batch. For BERT, the subphrase pairs occupy their original positions in the sentence, while we pad and apply an attention mask over all other positions. For the BiLSTM, we align them to the left, with [SEP] in between the two parts of the pair.

We performed an ablation experiment to test whether removing the augmented examples would affect BERT's performance. Using the same grid-search setting, we see that BERT's dev set accuracy decreased from 88.25% to 55.42%, and test set accuracy decreased from 88.50% to 54.51%. This indeed shows that the above data augmentation method is important for BERT to learn the type of generalization required for the hard MQNLI task.

## B.3 Training Procedure

For the BiLSTM, we use 256 dimensions for token embeddings and 128 dimensions for the hidden states in each LSTM direction. We grid search for $\{2, 4, 6\}$ layers. We randomly initialize each element in the token embeddings from the distribution $\mathcal{N}(0, 1)$ scaled down by a factor of 0.1. We use a batch size of $768 = 64 \times 12$, with 64 original examples per batch and 11 augmented examples for each one. We apply a dropout of 0.1, and grid search for learning rates in $\{0.001, 0.0001\}$. We train for a maximum of 400 epochs and perform early stopping when the dev set accuracy does not increase for 20 epochs. We train each grid search setting 3 times with different random seeds.

For BERT, we use the same model architecture for the uncased base variant. We use a batch size of $192 = 16 \times 12$, and grid search for learning rates in $\{2.0 \times 10^{-5}, 5.0 \times 10^{-5}\}$. We train for a maximum of $\{3, 4\}$ epochs. We warm up the learning rate linearly from 0 to the specified value in the first 25% of steps of the first epoch, and linearly decrease the learning rate to 0 following that until the end of training.

All models were trained with 1 GPU core on a cluster with models including GeForce RTX 2080 Ti, GeForce GTX Titan X, Titan XP and Titan V, each with 11-12GB memory. Each instance of the grid search took on average 5.5 hours to train. We repeated each grid search setting with 4 different random seeds and took the instance with the highest dev set accuracy.

## B.4 Interchange experiment details

There are 14 intermediate nodes in the high-level causal model (NegP, $QP_{Obj}$, $Q_{Subj}$, $NP_{Subj}$, $Adj_{Subj}$, $N_{Subj}$, Neg, VP, Adv, V, $Q_{Obj}$, $NP_{Obj}$, $Adj_{Obj}$, $N_{Obj}$). For each high-level node, we conducted a set of interchange experiments on each one of 11 BERT layers (excluding the final layer, since only the [CLS] token causally impacts the output). Each high-level node has its own fixed set of hand-specified intervention locations in the time-step/sentence length dimension, and we use the same intervention locations on each layer. For each of the $14 \times 11 = 154$ interchange experiments, it took on average 1.15 hours to run using the same computation resources mentioned above.

# C  Probing Details

## C.1  Probe Models

Our probe models are single-layer softmax classifiers: $y_i \propto \text{softmax}(Ah_i + b)$ where $h_i$ is a hidden representation and $y_i \in \mathbb{R}$. Following Hewitt and Liang [12], to control the dimensionality of $A$, we factorize it in the form $A = LR$ where $L \in \mathbb{R}^{|\mathcal{R}| \times \ell}$ and $R \in \mathbb{R}^{\ell \times d}$ where $d$ is the dimensionality of $h_i$.

We train the probes on hidden representations of a set of 12,800 examples that are randomly selected from the model's original training set. We additionally take 2,000 examples to form a development set for early stopping. We filter out examples for which the model outputs a wrong prediction.

For training, we perform a grid search, maximizing for selectivity. We set a dropout of 0.1, and apply early stopping when the development set loss does not increase for 4 epochs. We train for a maximum of 40 epochs. We also anneal the learning rate by a factor of 0.5 if the dev set loss did not increase in the last epoch. We use a batch size of 512, learning rates in $\{0.001, 0.01\}$, weight decay regularization constants in $\{0.01, 0.1\}$. We set $\ell \in \{8, 32\}$ for restricting the maximum rank of the linear matrix $A$.

Using the same computation resources described above, each grid search setting took approximately 5 hours to run. For each grid search setting we trained a separate probe for every possible $\langle$causal model node, BERT representation$\rangle$ combination, where for the latter we use the intervention locations outlined in the "Alignment Search" part of Section 5.1 on each BERT layer.

## C.2  Control Task

For each high-level node $N$, we construct a random mapping $\text{Control}_N : \mathcal{S}_N \mapsto \mathcal{L}_N$ where $\mathcal{S}_N$ is the set of all aligned subexpressions under the node $N$ and $\mathcal{L}_N$ is the output label space. For phrasal nodes (VP, NegP, etc.) and aligned verbs and nouns, $\mathcal{L}_N$ is the set of 7 possible relations $\{\#, \equiv, \sqsubset, \sqsupset, |, \smallfrown, \smallsmile\}$ from MacCartney and Manning [17]. For aligned quantifiers, the label space is the set of all projectivity signatures that can be produced by their composition.

Similar to Hewitt and Liang [12], $\text{Control}_N$ will assign the same control label regardless of the context as long as its input consists of the same tokens. Consequently, the possible input space $\mathcal{S}_N$ grows exponentially larger if $N$ corresponds to longer subphrases (such as NegP and $\text{QP}_{\text{Obj}}$), and the control task becomes much more difficult to solve, resulting in near random accuracies.

## C.3  Extended Probe Analysis

In Figures 5–7 we report some more representative selectivity and accuracy results for our probing experiments on BERT trained on the hard variant, juxtaposed against intervention experiments on the same model. For open-class words and full phrases, probing and intervention show similar trends. For aligned closed-class words, we find near-zero selectivity because the domain of the control function is so small.

In general, probing and intervention experiments for relations between aligned single open-class words (i.e., $\text{N}_{\text{Subj}}$, $\text{Adj}_{\text{Subj}}$, $\text{N}_{\text{Obj}}$, $\text{Adj}_{\text{Obj}}$, Adv, V) show similar trends, which can be seen in Figures 4c–4b. Every location except those above the [CLS] and [SEP] tokens has a near-100% accuracy, while selectivity is only high in the last few layers. Lower layers of BERT contains more information about word identity and hence may allow the probe to memorize each input pair, resulting in higher control task accuracy and lower selectivity for lower layers.

Probing experiments for relations between aligned multi-word subphrases (i.e., $\text{NP}_{\text{Subj}}$, VP, $\text{NP}_{\text{Obj}}$, $\text{QP}_{\text{Obj}}$ and NegP) show similar trends as shown in the row of figures 6m to 6h. As described in Section C.2, all control probes for these achieve near-random performance, so selectivity and accuracy differ by the random baseline accuracy, which is evident by comparing figures 6m and 6n.

On the other hand, probing experiments for aligned closed-class words (quantifiers and negation) have near-zero selectivity, as shown in Figure 6a. This is because the domain of the control function is the small set of closed-class word pairs, so memorizing the identity of these words becomes trivial for the probe.

# D  Probing and Intervention Heatmaps

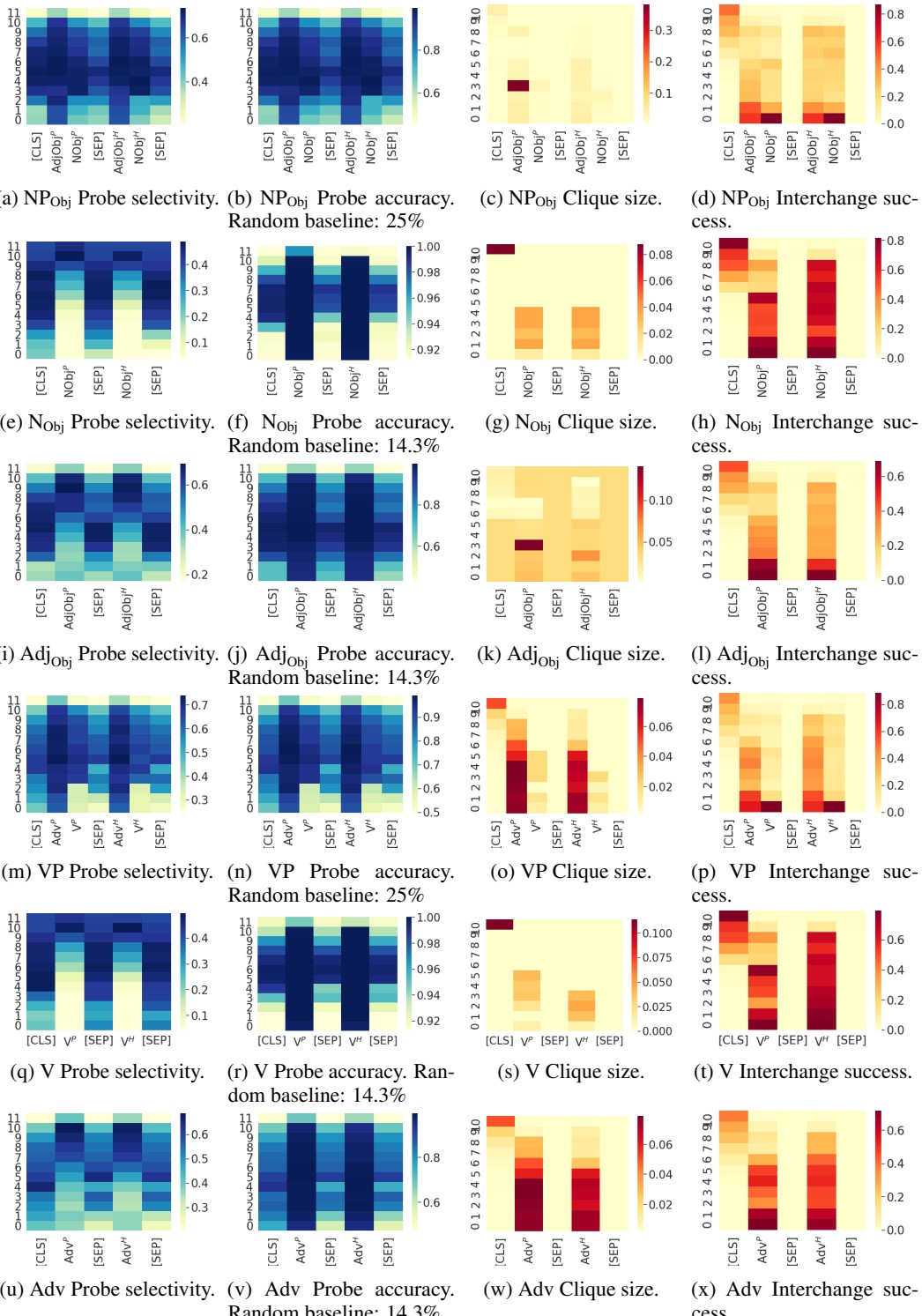

(a) NP$_{Obj}$ Probe selectivity.  (b) NP$_{Obj}$ Probe accuracy. Random baseline: 25%  (c) NP$_{Obj}$ Clique size.  (d) NP$_{Obj}$ Interchange success.

(e) N$_{Obj}$ Probe selectivity.  (f) N$_{Obj}$ Probe accuracy. Random baseline: 14.3%  (g) N$_{Obj}$ Clique size.  (h) N$_{Obj}$ Interchange success.

(i) Adj$_{Obj}$ Probe selectivity.  (j) Adj$_{Obj}$ Probe accuracy. Random baseline: 14.3%  (k) Adj$_{Obj}$ Clique size.  (l) Adj$_{Obj}$ Interchange success.

(m) VP Probe selectivity.  (n) VP Probe accuracy. Random baseline: 25%  (o) VP Clique size.  (p) VP Interchange success.

(q) V Probe selectivity.  (r) V Probe accuracy. Random baseline: 14.3%  (s) V Clique size.  (t) V Interchange success.

(u) Adv Probe selectivity.  (v) Adv Probe accuracy. Random baseline: 14.3%  (w) Adv Clique size.  (x) Adv Interchange success.

Figure 5: Full probing and interchange intervention results for high-level nodes NP$_{Obj}$, N$_{Obj}$, Adj$_{Obj}$, VP, V, and Adv. Vertical axes denote BERT layers and horizontal axes denote the token position of hidden representations. Intervention success rates are based on experiments with a change in the output label. Clique sizes are reported as a percentage of all examples.

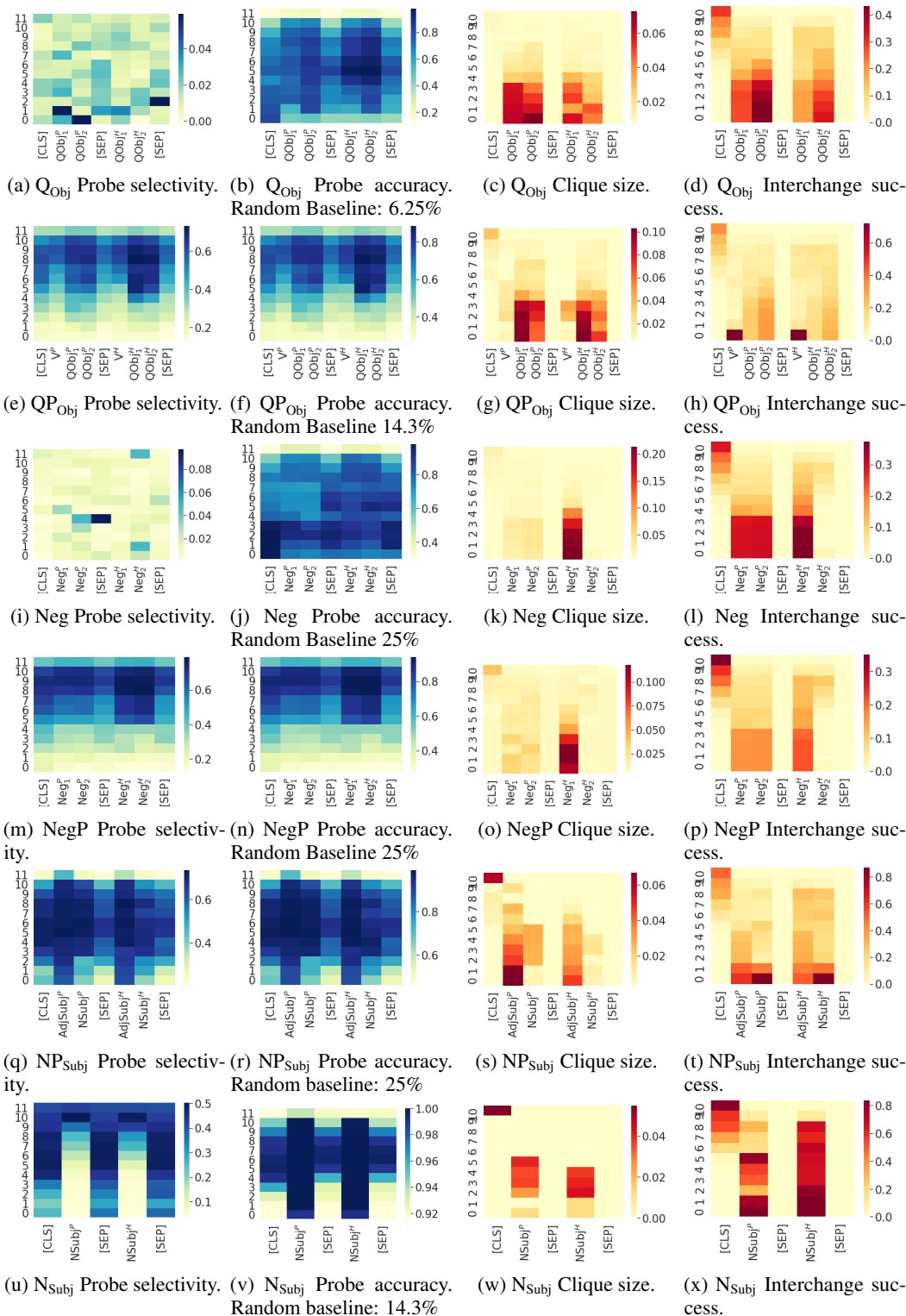

Figure 6: Full probing and interchange intervention results on the high level nodes $Q_{Obj}$, $QP_{Obj}$, Neg, NegP, $NP_{Subj}$ and $N_{Subj}$. Vertical axes denote BERT layers and horizontal axes denote the token position of hidden representations. Intervention success rates are based on experiments with a change in the output label. Clique sizes are reported as a percentage of all examples.

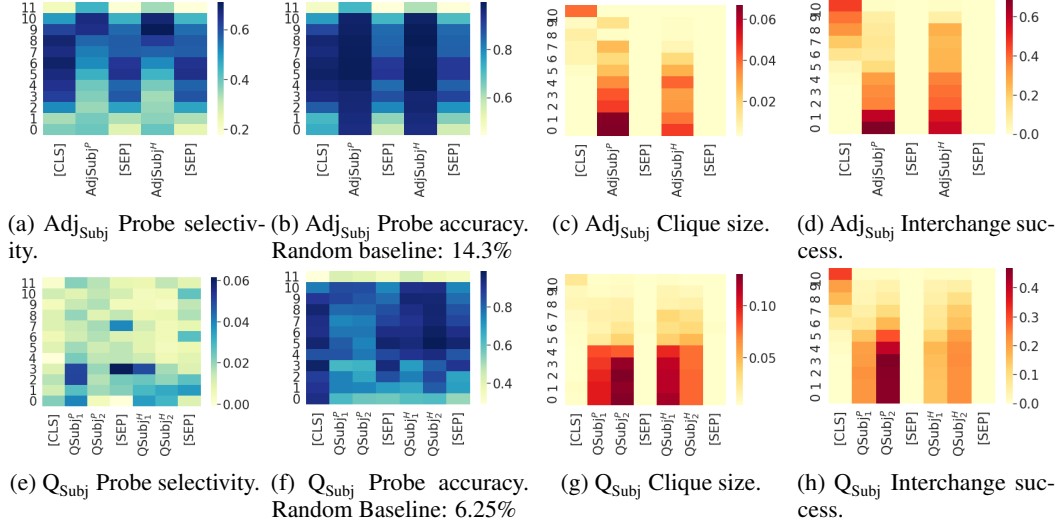

(a) Adj$_{Subj}$ Probe selectivity.

(b) Adj$_{Subj}$ Probe accuracy. Random baseline: 14.3%

(c) Adj$_{Subj}$ Clique size.

(d) Adj$_{Subj}$ Interchange success.

(e) Q$_{Subj}$ Probe selectivity.

(f) Q$_{Subj}$ Probe accuracy. Random Baseline: 6.25%

(g) Q$_{Subj}$ Clique size.

(h) Q$_{Subj}$ Interchange success.

Figure 7: Full probing and interchange intervention results for high-level nodes Adj$_{Subj}$ and Q$_{Subj}$. Vertical axes denote BERT layers and horizontal axes denote the token position of hidden representations. Intervention success rates are based on experiments with a change in the output label. Clique sizes are reported as a percentage of all examples.

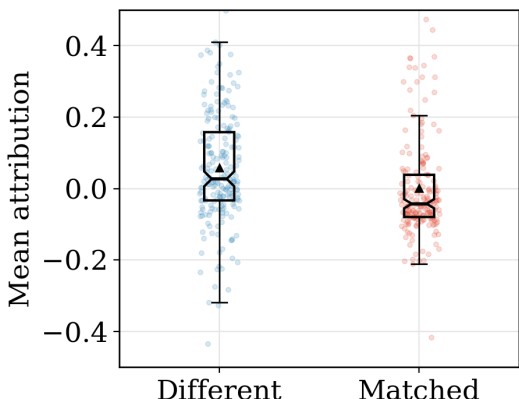

Figure 8: Integrated Gradients values for examples in which the premise and hypothesis differ by in exactly one aligned position. 'Different' refers to the IG value for this position, and 'Matched' is a randomly selected different position from each example. The two populations are different according to a Wilcoxon signed-rank test ($p < 0.00001$). The 'Different' positions have positive attribution on average, aligning with our expectation that they tend to be decisive for the output prediction.

# E   Integrated Gradients

We report attributions for the first BERT layer; later layers tend to concentrate importance onto the [CLS] token, since it is the direct basis for the classifier head in our model. To simplify the analysis, we restrict attention to examples in which exactly one position is different across the premise and hypothesis, and 'Matched' is a randomly selected position from elsewhere in the example. We see that the 'Matched' are positive in general, which aligns with our expectation that they are the most important positions in these examples (Figure 8).

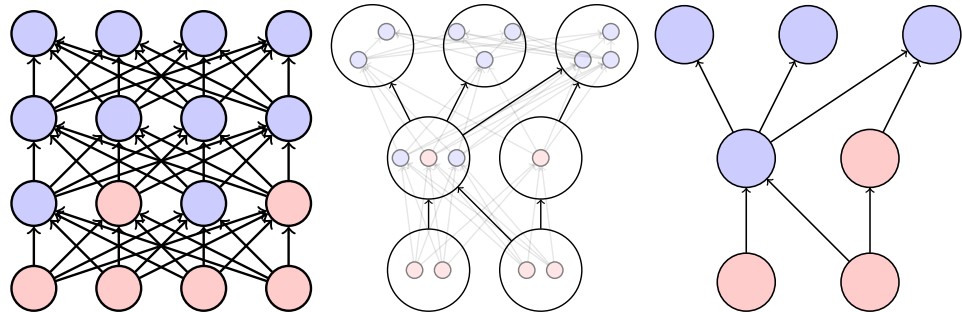

Figure 9: Schematic depicting constructive abstraction [1]. The variables of the low-level model (left) are divided into partitions (center) such that each low-level partition corresponds to a high level variable from the high-level model (right). The circles represent variables and the arrows represent causal dependencies. Blue circles are variables that are not being intervened on and red circles are variables that are being intervened on. Observe that a low-level causal dependence between partitions does not necessarily result in a high-level causal dependence between variables and that not every low-level intervention results in a high level intervention.

# F Background on Causal Models and Causal Abstraction

In this appendix we provide relevant background on causal models and causal abstraction, sufficient to define the notion of *constructive abstraction*.

## F.1 Causal Models

**Definition F.1.** (Signatures) A signature $S$ is a pair $(\mathcal{V}, \mathcal{R})$, where $\mathcal{V}$ is a set of variables and $\mathcal{R}$ is a function that associates with every variable $X \in \mathcal{V}$ a nonempty set $\mathcal{R}(X)$ of possible values. If $\mathbf{X} = (X_1, \ldots, X_n)$, $\mathcal{R}(\mathbf{X})$ denotes the cross product $\mathcal{R}(X_1) \times \cdots \times \mathcal{R}(X_n)$.

**Definition F.2.** (Causal models) A causal model $M$ is a pair $(\mathcal{S}, \mathcal{F})$, where $\mathcal{S}$ is a signature and $\mathcal{F}$ defines a function that associates with each variable $X$ a structural equation $\mathcal{F}^X$ giving the value of $X$ in terms of the values of other variables. Formally, the equation $\mathcal{F}^X$ maps $\mathcal{R}(\mathcal{V} - \{X\})$ to $\mathcal{R}(X)$, so $\mathcal{F}^X$ determines the value of $X$, given the values of all the other variables in $\mathcal{V}$.

**Definition F.3.** (Dependence) $X$ causes $Y$ according to $M$, denoted $X \rightsquigarrow Y$, if there is some setting of the variables other than $X$ and $Y$ such that varying the value of $X$ results in a variation in the value of $Y$; that is, there is a setting $\mathbf{z}$ of the variables $\mathbf{Z} = \mathcal{V} - \{X, Y\}$ and values $x$ and $x'$ of $X$ $\mathcal{F}^Y(x, \mathbf{z}) \neq \mathcal{F}^Y(x', \mathbf{z})$.

**Definition F.4.** (Intervention) An intervention $i$ has the form $\mathbf{X} \leftarrow \mathbf{x}$, where $\mathbf{X}$ is a vector of variables. Intuitively, this means that the values of the variables in $\mathbf{X}$ are set to $\mathbf{x}$. Setting the value of some variables $\mathbf{X} \leftarrow \mathbf{x}$ in a causal model $M = (\mathcal{S}, \mathcal{F})$ results in a new causal model, denoted $i(M)$, which is identical to $M$, except that $\mathcal{F}$ is replaced by $i(\mathcal{F})$: for each variable $Y \notin \mathbf{X}$, $i(\mathcal{F}^Y) = \mathcal{F}^Y$ (i.e., the equation for $Y$ is unchanged), while for each $X' \in \mathbf{X}$, $i(F^{X'})$ is the constant function sending all arguments to $x'$ (where $x'$ is the value in $\mathbf{x}$ corresponding to $X_i$).

When we write out the structured equations for a variable $X$, for simplicity's sake, we treat $\mathcal{F}^X$ as a map from $\mathcal{R}(\{Y \in \mathcal{V} : Y \rightsquigarrow X\})$ to $\mathcal{R}(X)$.

Note that interventions $\mathbf{X} \leftarrow \mathbf{x}$ correspond 1–1 with variable settings $\mathbf{x}$. We make use of this in what follows.

## F.2 Constructive Abstraction

The following definitions are in agreement with the definitions from Beckers and Halpern [1], but differ somewhat in presentation. We additionally omit exogenous variables, as they play no role in our deterministic setting. In this section we take causal models to be pairs $(M, \mathcal{I})$, with a set $\mathcal{I}$ of *admissible interventions* made explicit.

**Definition F.5.** (Projection and Inverse Projection) Given some $\mathbf{v} \in \mathcal{R}(\mathcal{V})$ and $\mathbf{X} \subseteq \mathcal{V}$, define $\mathsf{Proj}(\mathbf{v}, \mathbf{X})$ to be the restriction of $\mathbf{v}$ to the variables in $\mathbf{X}$. Given some $\mathbf{x} \subseteq \mathcal{V}(\mathbf{X})$, the inverse

$\mathsf{Proj}^{-1}(\mathbf{x})$ is defined as usual:

$$\{\mathbf{v} \in \mathcal{R}(\mathcal{V}) : \mathbf{x} \text{ is the restriction of } \mathbf{v} \text{ to } \mathbf{X}\}.$$

We are interested in (possibly partial) functions $\tau : \mathcal{R}_L(\mathcal{V}_L) \to \mathcal{R}_H(\mathcal{V}_H)$ mapping settings of low-level variables to settings of high-level variables. Such a function $\tau$ naturally induces a function $\omega_\tau$ between sets of interventions, where $\omega_\tau(\mathbf{x}) = \mathbf{y}$ just in case

$$\tau(\mathsf{Proj}^{-1}(\mathbf{x})) = \mathsf{Proj}^{-1}(\mathbf{y}).$$

We are now in a position to define $\tau$-abstraction:

**Definition F.6.** ($\tau$-abstraction) Fix a function $\tau : \mathcal{R}_L(\mathcal{V}_L) \to \mathcal{R}_H(\mathcal{V}_H)$, which in turn fixes $\omega_\tau : \mathcal{I}_L \to \mathcal{I}_H$. We say $(M_H, \mathcal{I}_H)$ is a $\tau$-abstraction of $(M_L, \mathcal{I}_L)$ if the following hold:

1. $\tau$ is surjective.

2. $\omega_\tau$ is surjective.

3. for all $i_L \in \mathcal{I}_L$ we have $\tau(i_L(M_L)) = \omega_\tau(i_L)(M_H)$.

One way to think of this is: $\tau$ is a map from $\mathcal{R}(\mathcal{V}_L)$ to $\mathcal{R}(\mathcal{V}_H)$, which in turn induces a map $\omega_\tau$ from the space of *projections* on $\mathcal{R}(\mathcal{V}_L)$ to *projections* on $\mathcal{R}(\mathcal{V}_H)$. The conditions on $\tau$-abstraction below then simply become that $\tau$ and $\omega_\tau$ are both total and surjective on their respective (co)domains, and a second condition that can be easily encoded in terms of potential outcomes. For any setting/projection $\mathbf{x}$ at the low-level, we require that $M_L \models \mathbf{v}_\mathbf{x}$ iff $M_H \models \tau(\mathbf{v})_{\omega_\tau(\mathbf{x})}$.

Finally, to be a *constructive* $\tau$-abstraction we simply require that $\tau$ decompose into a family of "component" functions, as below.

**Definition F.7** (Constructive $\tau$-abstraction). $(M_H, \mathcal{I}_H)$ is a constructive $\tau$-abstraction of $(M_L, \mathcal{I}_L)$ if, in addition to being a $\tau$-abstraction, we can associate with each $X_H$ a subset $P_{X_H}$ of $\mathcal{V}_L$, such that the mapping $\tau : \mathcal{R}(\mathcal{V}_L) \to \mathcal{R}(\mathcal{V}_H)$ decomposes into a family of functions $\tau_{X_H} : \mathcal{R}(P_{X_H}) \to \mathcal{R}(X_H)$. We say $M_H$ is a constructive abstraction of $M_L$ if it is a constructive $\tau$-abstraction for some $\tau$.

In other words, for a constructive abstraction it suffices to define the component functions $\tau_{X_H}$, as these completely determine $\tau$. In fact, the maps $\tau_{X_H}$ more generally induce a (partial) function from projections of $\mathcal{R}(\mathcal{V}_L)$ to (in fact, onto) projections of $\mathcal{R}(\mathcal{V}_H)$ in the following sense. For any setting $\mathbf{h} = [h_1 \ldots h_k]$ of high-level variables $H_1, \ldots, H_k$ we can find low-level setting $\mathbf{y}$ such that projections of $\mathbf{y}$ map via $\tau_{H_i}$ to $h_i$. Slightly abusing notation, denote this (partial) low-level setting $\mathbf{y}$ as $\tau^{-1}(\mathbf{h})$. So, in particular when $\mathbf{h}$ corresponds to an intervention in $\mathcal{I}_H$, the setting $\tau^{-1}(\mathbf{h})$ should specify a corresponding intervention in $\mathcal{I}_L$. Indeed, point (2) of Def. F.6 tells us that (the intervention corresponding to) $\tau^{-1}(\mathbf{h})$ should be mapped via $\omega_\tau$ to (the intervention corresponding to) $\mathbf{h}$.

## G   Causal Abstraction Analysis of $C_+$

### G.1   Formal Definition of $C_+$

We define the causal model $C_+ = (\mathcal{V}_+, \mathcal{R}_+, \mathcal{F}_+)$ as follows (where $\mathbb{N}_k = \{0, \ldots, k\}$):

$$\mathcal{V}_+ = \{X, Y, Z, W, S_1, S_2\}$$

$$\mathcal{R}_+(V) = \mathbb{N}_9, \text{ for } V \in \{X, Y, Z, W\}$$
$$\mathcal{R}_+(S_1) = \mathbb{N}_{18}$$
$$\mathcal{R}_+(S_2) = \mathbb{N}_{27}$$

$$\mathcal{F}_+^X = \mathcal{F}_+^Y = \mathcal{F}_+^Z = 0$$
$$\forall z \in \mathcal{R}(Z) : \mathcal{F}_+^W(z) = z$$
$$\forall (x, y) \in \mathcal{R}(X) \times \mathcal{R}(Y) : \mathcal{F}_+^{S_1}(x, y) = x + y$$
$$\forall (s_1, w) \in \mathcal{R}(S_1) \times \mathcal{R}(W) : \mathcal{F}_+^{S_2}(s_1, w) = s_1 + w$$

## G.2 Formal Definition of $N_+$

In the main text, we did not provide a specific identity for $N_+$. Here, we define $N_+$ to be a feed forward network, which we represent directly as a causal model $C_{N_+} = (\mathcal{V}_{N_+}, \mathcal{R}_{N_+}, \mathcal{F}_{N_+})$. The location $L_1$ from Figure 1 is the hidden unit $H_3$, the location $L_2$ is the hidden unit $H_1$.

Let $W \in \mathbb{R}^{30 \times 3}$; for $k \in \{1, 3\}$ let $W_{jk} = j \bmod 10$ if $0 \leq j \leq 20$, otherwise $W_{jk} = 0$, and let $W_{j2} = 0$ if $0 \leq j \leq 20$, otherwise $W_{j2} = j \bmod 10$. Let $U \in \mathbb{R}^3$ and $U = [1, 1, 0]$.

$$\mathcal{V}_{N_+} = \{D_x, D_y, D_z, H_1, H_2, H_3, O\}$$

$$\mathcal{R}_{N_+}(D_x) = \mathcal{R}_{N_+}(D_y) = \mathcal{R}_{N_+}(D_z) = \{0, 1\}^{10}$$
$$\mathcal{R}_{N_+}(O) = \mathcal{R}_{N_+}(H_1) = \mathcal{R}_{N_+}(H_2) = \mathcal{R}_{N_+}(H_3) = \mathbb{R}$$

$$\mathcal{F}_{N_+}^{D_x} = \mathcal{F}_{N_+}^{D_y} = \mathcal{F}_{N_+}^{D_z} = 0$$
$$\forall \mathbf{x} \in \mathcal{R}_{N_+}(D_x) \times \mathcal{R}_{N_+}(D_y) \times \mathcal{R}_{N_+}(D_z) : [\mathcal{F}_{N_+}^{H_1}(\mathbf{x}), \mathcal{F}_{N_+}^{H_2}(\mathbf{x}), \mathcal{F}_{N_+}^{H_3}(\mathbf{x})] = \mathrm{ReLU}(\mathbf{x}W)$$
$$\forall \mathbf{h} \in \mathcal{R}_{N_+}(H_1) \times \mathcal{R}_{N_+}(H_2) \times \mathcal{R}_{N_+}(H_3) : \mathcal{F}_{N_+}^{O}(\mathbf{h}) = \mathrm{ReLU}(\mathbf{h}U)$$

This network uses one-hot representations $d_x, d_y, d_z \in \{0, 1\}^{10}$ to represent inputs from $\mathbb{N}_9$.

## G.3 Proving $C_+$ is an abstraction of $N_+$

We now prove that $C_+$ is an abstraction $C_{N_+}$

We define the mapping $\tau : \mathcal{R}_{N_+}(\mathcal{V}_{N_+}) \to \mathcal{R}_+(\mathcal{V}_+)$ as follows. We first partition the variables of $N_+$ into cells: $P_X = \{D_x\}$, $P_Y = \{D_y\}$, $P_Z = \{D_z\}$, $P_W = \{H_1\}$, $P_{S_1} = \{H_3\}$, $P_{S_2} = \{O\}$, $P_\emptyset = \{H_2\}$. To define $\tau$ it suffices to define the component functions $\tau_V$ for $V \in \mathcal{V}_+$. Let $B : \{0, 1\}^{10} \to \mathbb{N}_9$ be the partial function s.t. $B([v_1, v_2, \ldots, v_{10}]) = k$ if $v_k = 1$ and $v_j = 0$ for $j \neq k$. Set $\tau_X, \tau_Y, \tau_Z$ all equal to $B$, and let $\tau_W, \tau_{S_1}, \tau_{S_2}$ all be the identity function.

Let $\mathcal{I}_+$ be the set of all interventions on $C_+$ that determine values for (at least) $X$, $Y$, and $Z$. Let $\mathcal{I}_{N_+} = dom(\omega_\tau)$. That is, $\mathcal{I}_{N_+}$ includes exactly the (interventions corresponding to) projections of $\mathcal{R}_{N_+}(\mathcal{V}_{N_+})$ that map via $\omega_\tau$ to some admissible intervention on $C_+$. Because elements of $\mathcal{I}_+$ always determine values for $X, Y, Z$, every intervention in $\mathcal{I}_{N_+}$ determines a value for each of $D_x, D_y, D_z$. In fact, these values are guaranteed to be in the domains of $\tau_X, \tau_Y, \tau_Z$, respectively.

We now prove the three conditions guaranteeing $(C_+, \mathcal{I}_+)$ is a $\tau$-abstraction of $(C_{N_+}, \mathcal{I}_{N_+})$.

(1) The first point is that the map $\tau$ is surjective. Take an arbitrary $(x, y, z, w, s_1, s_2) \in \mathcal{R}_+(\mathcal{V}_+)$. We determine an element of $\mathcal{R}_{N_+}(\mathcal{V}_{N_+})$ as follows: $[d_x d_y d_z] = B^{-1}([x, y, z])$, $[h_1 h_2 h_3] = [s_1 d_2 s_1]$, and $o = s_2$. It's then clear that $\tau(d_x, d_y, d_z, h_1, h_2, h_3, o) = (x, y, z, w, s_1, s_2)$. As $(x, y, z, w, s_1, s_2)$ was chosen arbitrarily, $\tau$ is surjective.

(2) The second point is that $\omega_\tau$ must also surject onto the set $\mathcal{I}_+$ of all interventions on $C_+$. Any intervention $i_+ \in \mathcal{I}_+$ can be identified with a vector $\mathbf{i}^+$ of values of variables in $\mathcal{V}_+$. By definition of $\mathcal{I}_+$, $i_+$ fixes at least the values of $X, Y, Z$. Consider the intervention $i_{N_+}$ that sets $D_x$, $D_y$, and $D_z$ to the one-hot representations of $X$, $Y$, and $Z$ for the values they were set. Furthermore, if $i_+$ sets $W$ to $w$ then $i_{N_+}$ sets $H_1$ to $w$ and if $i_+$ sets $S_2$ to $s_2$, then $i_{N_+}$ sets $H_3$ to $s_2$. It suffices to show that $\omega_\tau(i_{N_+}) = i_+$. In other words, we need to show that $\tau(\mathsf{Proj}^{-1}(\mathbf{i}^{N_+})) = \mathsf{Proj}^{-1}(\mathbf{i}^+)$.

First, we show for all $\mathbf{v}_L \in \mathsf{Proj}^{-1}(\mathbf{i}^{N_+})$ that $\tau(\mathbf{v}_L) \in \mathsf{Proj}^{-1}(\mathbf{i}^+)$. By construction of $i_+$, any variables fixed by $i_{N_+}$ will correspond (via $\tau$ component functions) to values of variables fixed by $i_+$, except for the variable $H_3$, which has no corresponding high level variable. We merely need to observe that for any values of variables *not* set by $i_{N_+}$, there exist corresponding values for the variables that are *not* set by $i_+$, such that the appropriate $\tau$ component functions map the former to the latter (with the exception of $H_3$, which has no corresponding high level variable). This is obvious from the definition of the components of $\tau$.

Second, we show for all $\mathbf{v}_H \in \mathsf{Proj}^{-1}(\mathbf{i}^+)$ there is $\mathbf{v}_L \in \mathsf{Proj}^{-1}(\mathbf{i}^{N_+})$ such that $\tau(\mathbf{v}_L) = \mathbf{v}_H$. Again, by construction of $i_+$, any variables fixed by $i_+$ will correspond (via $\tau$ component functions) to values of variables fixed by $i_{N_+}$. We merely need to observe that for any values of variables *not* set by $i_+$, there exist corresponding values for the variables *not* set by $i_{N_+}$, such that the appropriate $\tau$

component functions map the former to the latter, with $H_3$ taking on any value. This is obvious from the definition of the components of $\tau$. This concludes the argument that $\omega_\tau(i_{N_+}) = i_+$.

(3) Finally, we need to show for each $i_{N_+} \in dom(\omega_\tau)$ that $\tau(i_{N_+}(C_{N_+})) = \omega_\tau(i_{N_+})(C_+)$. The point here is that the two causal processes unfold in the same way, under any intervention.

Indeed, pick any $i_{N_+}$ and suppose that $i_+ = \omega_\tau(i_{N_+})$. We know that $i_+$ fixes values $x, y, z$ of $X, Y, Z$, and likewise that $i_{N_+}$ fixes values $d_x, d_y, d_z$ of $D_x, D_y, D_z$ such that $\tau_{D_j}(x_j) = d_j$ for $j \in \{1, 2, 3\}$. Any other variables fixed by $i_+$ from among $W, S_1, S_2$ will likewise correspond (via $\tau_W, \tau_{S_1}, \tau_{S_2}$) to values of $H_2, H_1, O$ fixed by $i_{N_+}$. We merely need to observe that any variables that are *not* set by $i_+$ and $i_{N_+}$ will still correspond via the appropriate $\tau$-component, given their settings in $i_+(C_+)$ and $i_{N_+}(C_{N_+})$. The mechanisms in $C_{N_+}$ were devised precisely to guarantee this.

Thus we have fulfilled the three requirements and we have shown that $C_+$ is an abstraction $C_{N_+}$.

The proof that $C_{NatLog}$ is a constructive abstraction of $N_{NLI}$ follows this same pattern.

# H  Causal Abstraction Analysis of $C_{NatLog}$

## H.1  Formal Definition of $C_{NatLog}$

We formally define the model $C_{NatLog} = (\mathcal{V}_{NatLog}, \mathcal{R}_{NatLog}, \mathcal{F}_{NatLog})$ as follows:

$$\mathcal{V}_{NatLog} = \left\{ \begin{array}{c} Q^P_{Subj}, Q^H_{Subj}, Neg^P_{Subj}, Neg^H_{Subj}, N^P_{Subj}, N^H_{Subj}, Neg^P, Neg^H, Adv^P, Adv^H, \\ V^P, V^H, Q^P_{Obj}, Q^H_{Obj}, Neg^P_{Obj}, Neg^H_{Obj}, N^P_{Obj}, N^H_{Obj}, Q_{Subj}, Neg_{Subj}, N_{Subj}, Neg, Adv \\ Q_{Obj}, Neg_{Obj}, N_{Obj}, NP_{Subj}, VP, NP_{Obj}, QP_{Obj}, NegP, QP_{Subj} \end{array} \right\}$$

$$\mathcal{R}_{NatLog}(Q^P_{Subj}) = \mathcal{R}_{NatLog}(Q^H_{Subj}) = \mathcal{R}_{NatLog}(Q^H_{Subj}) = \mathcal{R}_{NatLog}(Q^H_{Subj})$$
$$= \{no,\ some,\ every,\ not\ every\}$$
$$\mathcal{R}_{NatLog}(Neg^P) = \mathcal{R}_{NatLog}(Neg^H) = \{not, \epsilon\}$$
$$\mathcal{R}_{NatLog}(Neg^P_{Subj}) = \mathcal{R}_{NatLog}(Neg^H_{Subj}) = Neg_{Subj}$$
$$\mathcal{R}_{NatLog}(N^P_{Subj}) = \mathcal{R}_{NatLog}(N^H_{Subj}) = \mathbf{N}_{Subj}$$
$$\mathcal{R}_{NatLog}(Adv^P) = \mathcal{R}_{NatLog}(Adv^H) = Adv_{Subj}$$
$$\mathcal{R}_{NatLog}(V^P) = \mathcal{R}_{NatLog}(V^H) = \mathbf{V}_{Subj}$$
$$\mathcal{R}_{NatLog}(Neg^P_{Obj}) = \mathcal{R}_{NatLog}(Neg^H_{Obj}) = Neg_{Obj}$$
$$\mathcal{R}_{NatLog}(N^P_{Obj}) = \mathcal{R}_{NatLog}(N^H_{Obj}) = \mathbf{N}_{Obj}$$
$$\mathcal{R}_{NatLog}(Q_{Obj}) = \mathcal{R}_{NatLog}(Q_{Subj}) = \mathcal{Q}$$
$$\mathcal{R}_{NatLog}(Neg) = \mathcal{N}$$
$$\mathcal{R}_{NatLog}(Neg_{Obj}) = \mathcal{R}_{NatLog}(Neg_{Subj}) = \mathcal{R}_{NatLog}(Adv) = \mathcal{A}$$
$$\mathcal{R}_{NatLog}(N_{Obj}) = \mathcal{R}_{NatLog}(N_{Subj}) = \mathcal{R}_{NatLog}(V) = \{\#, \equiv\}$$
$$\mathcal{R}_{NatLog}(NP_{Subj}) = \mathcal{R}_{NatLog}(NP_{Obj}) = \mathcal{R}_{NatLog}(VP) = \{\#, \equiv, \sqsubset, \sqsupset\}$$
$$\mathcal{R}_{NatLog}(QP_{Obj}) = \mathcal{R}_{NatLog}(NegP) = \mathcal{R}_{NatLog}(QP_{Subj}) = \{\#, \equiv, \sqsubset, \sqsupset, |, \char94, \smile\}$$

$$\mathcal{F}_N = COMP \text{ for } N \in \{VP, NP_{Subj}, NP_{Obj}, NegP, QP_{Obj}, QP_{Subj}\}$$
$$\mathcal{F}_N = REL \text{ for } N \in \{V, N_{Subj}, N_{Obj}\}$$
$$\mathcal{F}_N = PROJ \text{ for } N \in \{Q_{Obj}, Q_{Subj}, Adv, Neg_{Subj}, Neg_{Obj}, Neg\}$$

The set $\{\#, \equiv, \sqsubset, \sqsupset, |, \char94, \smile\}$ contains the seven relations used in the natural logic of MacCartney and Manning [17]. The set $\mathbf{N}_{Subj}$ contains the subject nouns used to create MQNLI, $\mathbf{N}_{Obj}$ the set of object nouns, $\mathbf{Adj}_{Subj}$ the subject adjectives, $\mathbf{Adj}_{Obj}$ the object adjectives, $\mathbf{V}$ the verbs, and Adv the adverbs. Additionally, $\mathcal{Q}$ is the set of joint projectivity signatures between *every*, *some*, *not every*, and *no*, $\mathcal{N}$ is the set of joint projectivity signatures between *not* and $\epsilon$, $\mathcal{A}$ is the set of joint projectivity signatures between intersective adjectives and adverbs and $\epsilon$. $REL(x, y)$ outputs the lexical relation

between $x$ and $y$. Finally, $\text{COMP}(f, x_1, x_2, \ldots, x_n) = f(x_1, x_2, \ldots, x_n)$ and $\text{PROJ}(f, g) = P_{f/g}$ where $P_{f/g}$ is the joint projectivity signature between $f$ and $g$. See Geiger et al. [10] for details about these sets and functions.

## H.2  Formal Definition of $C_{NatLog}^N$

For some non-leaf node $N$ of the tree in Figure 2a, we define $C_{NatLog}^N$ to be the marginalization of $C_{NatLog}$ where all variables are removed other than the input variables

$$\mathcal{V}_{NatLog}^{Input} = \text{Q}_{Subj}^P, \text{Q}_{Subj}^H, \text{Neg}_{Subj}^P, \text{Neg}_{Subj}^H, \text{N}_{Subj}^P, \text{N}_{Subj}^H, \text{Neg}^P, \text{Neg}^H, \text{Adv}^P, \text{Adv}^H, \text{V}^P, \text{V}^H,$$

$$\text{Q}_{Obj}^P, \text{Q}_{Obj}^H, \text{Neg}_{Obj}^P, \text{Neg}_{Obj}^H, \text{N}_{Obj}^P, \text{N}_{Obj}^H$$

along with the output variable $\text{QP}_{Subj}$ and the intermediate variable $N$. For a definition of marginalization, see Bongers et al. [4].

## H.3  Formal definition of $N_{NLI}$

In the main text, $N_{NLI}$ could represent either our BERT model or our LSTM model. We will maintain this ambiguity, because while these two models are drastically different at the highest level of detail, for the sake of our analysis we can view them both as creating a grid of neural representations where each representation in the grid is caused by all representations in the previous row and causes all representations in the following row. We will now formally define the causal model $C_{N_{NLI}}$.

$$\mathcal{V}_{N_{NLI}} = \{R_{11}, R_{12}, \ldots, R_{1m}, \ldots R_{nm}, O\}$$

For the LSTM model $n = 2$ and for the BERT model $n = 12$. $m$ is the number of tokens in a tokenized version of an MQNLI example.

$$\mathcal{R}_{N_{NLI}}(R_{jk}) = \mathbb{R}^d \quad \mathcal{R}_{N_{NLI}}(O) = \{\text{entailment, contradiction, neutral}\}$$

For all $j$ and $k$ and where $d$ is the dimension of the vector representations.

$$\forall (r_{(j-1)1}, r_{(j-1)2}, \ldots, r_{(j-1)m}) \in \mathcal{R}_{N_{NLI}}(R_{(j-1)1} \times R_{(j-1)2} \times \cdots \times R_{(j-3)m})$$

$$\mathcal{F}_{N_{NLI}}^{R_{jk}}(r_{(j-1)1}, r_{(j-1)2}, \ldots, r_{(j-1)m}) = \mathbf{NN}_{jk}(r_{(j-1)1}, r_{(j-1)2}, \ldots, r_{(j-1)m})$$

where $\mathbf{NN}_{jk}$ is either the LSTM function or the BERT function that creates the neural representation at the $j$th row and $k$th column.

$$\forall r_{n1} \in \mathcal{R}_{N_{NLI}}(R_{n1}) \mathcal{F}_{N_{NLI}}^O(r_{n1}) = \mathbf{NN}_O(r_{n1})$$

where $\mathbf{NN}_O$ is the neural network that makes a three class prediction using the final representation of the [CLS] token.

See Appendix B for details about these functions.

## H.4  Proving $C_{NatLog}^N$ is an abstraction of $N_{NLI}$

We will now formally prove that that $C_{NatLog}^N$ is a constructive abstraction of $N_{NLI}$ if the following holds for all $e, e' \in \text{MQNLI}$, where the representation location $L$ is equivalent to the variable $R_{jk}$ for some $j$ and $k$. This would mean that every single one of our intervention experiments at this location are successful.

$$C_{NatLog}^{N \leftarrow e'}(e) = N_{NLI}^{L \leftarrow e'}(e) \tag{8}$$

We define the mapping $\tau : \mathcal{R}_{N_{NLI}}(\mathcal{V}_{N_{NLI}}) \to \mathcal{R}_{NatLog}^N(\mathcal{V}_{NatLog})$ as follows. We first partition the "low level" variables of $N_{NLI}$ into partition cells:

$$P_N = \{L\} \qquad\qquad P_{\text{QP}_{Subj}} = \{O\} \quad \forall X \in \mathcal{V}_{NatLog}^{Input}$$
$$P_X = \{R_{1j}, R_{1(j+1)}, \ldots, R_{1(j+k)}\}$$

where $R_{1j}, R_{1(j+1)}, \ldots, R_{1(j+k)}$ are the token vectors associated with the input variable $X$. Some of our causal model's input variables are tokenized into several tokens (see Appendix B for details).

To define $\tau$, it then suffices to define the component functions $\tau_V$ for each $V \in \mathcal{V}_{NatLog}$. Let $T : (\mathbb{R}^d)^+ \to \mathcal{V}_{NatLog}^{Input}$ be the partial function mapping sequences of token vectors to the input variable they correspond to, where $+$ is the Kleene plus operator. Let $P : \mathcal{R}^3 \to \{\text{entailment, neutral, contradiction}\}$ be the partial function mapping a vector of logits to the output prediction they correspond to. Finally, let $Q_L : \mathbb{R}^d \to \mathcal{R}_{NatLog}(N)$ be the partial function such that for all $e \in$ MQNLI, if $\mathbf{v}$ is the vector created by $N_{NLI}$ at location $L$ when processing input $e$ and $x$ is the value realized by $C_{NatLog}$ for the variable $N$ when processing input $e$, then $Q_L(\mathbf{v}) = x$.

For all $\forall X \in \mathcal{V}_{NatLog}^{Input}$, we set $\tau_X$ to be $T$. We additionally set $\tau_N$ to be $Q_L$ and $\tau_{\text{QP}_{Subj}}$ to be $P$.

Let $\mathcal{I}_{NatLog}$ be the set of all interventions on $C_{NatLog}$ that intervene on (i.e., determine the values for) at least the elements of $\mathcal{V}_{NatLog}^{Input}$. Let $\mathcal{I}_{N_{NLI}}$ be the set of interventions that is the domain of the partial function $\omega_\tau$. In other words, $\mathcal{I}_{N_{NLI}}$ includes exactly the projections of $\mathcal{R}_{N_{NLI}}(\mathcal{V}_{N_{NLI}})$ that map via $\omega_\tau$ to some intervention on $C_+$. The fact that $P, Q_L$, an $T$ are all proper partial functions prevent $\mathcal{I}_{N_{NLI}}$ from including all possible interventions on $C_{N_{NLI}}$.

We now prove the three conditions that must hold for $(C_{NatLog}, \mathcal{I}_{NatLog})$ to be a $\tau$-abstraction of $(C_{N_{NLI}}, \mathcal{I}_{N_{NLI}})$.

(1) The first point is to show the map $\tau$ is surjective. So take an arbitrary element $(\vec{v}^{input}, n, q) \in \mathcal{R}_{NatLog}(\mathcal{V}_{NatLog})$. We specify an element of $\mathcal{R}_{N_{NLI}}(\mathcal{V}_{N_{NLI}})$ as follows:

$$l = Q_L^{-1}(n) \qquad\qquad\qquad\qquad o = P^{-1}(q)$$

$$\forall v^{input} \in \vec{v}^{input} T^{-1}(v^{input}) = (r_{1j}, r_{1(j+1)}, \ldots, r_{1(j+k)})$$

where $r_{1j}, r_{1(j+1)}, \ldots, r_{1(j+k)}$ are the token vectors corresponding to the input variable $v^{input}$.

It's then patent that $\tau(r_{11}, \ldots, r_{n1}, r_{12}, \ldots r_{nm}, o) = (\vec{v}^{input}, n, q)$. As $(\vec{v}^{input}, n, q)$ was chosen arbitrarily, we have shown $\tau$ is surjective.

(2) The second point is that $\omega_\tau$ must also be surjective onto the set $\mathcal{I}_{NatLog}$ of interventions on $C_{NatLog}$. Any intervention $i_{NatLog} \in \mathcal{I}_{NatLog}$ can be identified with with a vector $\mathbf{i}^{NatLog}$ of values of variables in $\mathcal{V}_{NatLog}$. By the definition of $\mathcal{I}_{NatLog}$, $i_{NatLog}$ fixes the values of the variables in $\mathbf{V}^{input}$ and may also determine $N$ and/or QP$_{Subj}$. Consider the intervention $i_{N_{NLI}}$ corresponding to $\mathbf{i}^{\mathbf{N}_{NLI}} = \tau^{-1}(\mathbf{i}^{NatLog})$ as described in Section F.2. It suffices to show that $\omega_\tau(i_{N_{NLI}}) = i_{NatLog}$. In other words, we need to show parts 1, 2, and 3 from the definition above.

Part 1 is clear, since by the definition of $\mathcal{I}_{NatLog}$ we are guaranteed that $\mathbf{i}^{NatLog}$ determines values for $\mathbf{V}^{input}$, and hence $\mathbf{i}^{N_{NLI}}$ fixes values for $R_{11}, \ldots, R_{1m}$ in the domains of $\tau_{\mathbf{V}^{input}}$ for $V \in \mathbf{V}^{input}$. Then any intervention that intervenes only on the values of

Part 2 requires that for every $\mathbf{v}_{N_{NLI}} \in \text{Proj}^{-1}(\mathbf{i}^{N_{NLI}})$, we have $\tau(\mathbf{v}_{N_{NLI}}) \in \text{Proj}^{-1}(\mathbf{i}^{NatLog})$. Because of how we defined $i_{NatLog}$, any variables fixed by $i_{N_{NLI}}$ will correspond (via $\tau$ component functions) to values of variables fixed by $i_{NatLog}$, except for the variables $R_{jk} \notin \mathbf{V}^{input} \cup \{L\}$, which have no corresponding high level variables. We merely need to observe that, for any values for the variables that are *not* set by $i_{N_{NLI}}$, there exists corresponding values for the variables that are *not* set by $i_{NatLog}$ such that the appropriate $\tau$ component functions map the former to the latter, except for the variables $R_{jk} \notin \mathbf{V}^{input} \cup \{L\}$, which, again, have no corresponding high level variables. This is plainly obvious from the definition of the components of $\tau$.

Part 3 requires that for any $\mathbf{v}_{NatLog} \in \text{Proj}^{-1}(\mathbf{i}^{NatLog})$, there exists a $\mathbf{v}_{N_{NLI}} \in \text{Proj}^{-1}(\mathbf{i}^{N_{NLI}})$ such that $\tau(\mathbf{v}_{N_{NLI}}) = \mathbf{v}_{NatLog}$. Again, because of how we defined $i_{NatLog}$, any variables fixed by $i_{NatLog}$ will correspond (via $\tau$ component functions) to values of variables fixed by $i_{N_{NLI}}$. We merely need to observe that for any values for the variables that are *not* set by $i_{NatLog}$, there exists corresponding values for the variables that are *not* set by $i_{N_{NLI}}$, such that the appropriate $\tau$ component functions map the former to the latter, with $R_{jk} \notin \mathbf{V}^{input} \cup \{L\}$ taking on any value. This is plainly obvious from the definition of the components of $\tau$.

Thus, we have shown that $\omega_\tau(i_{N_{NLI}}) = i_{NatLog}$.

(3) Finally, we need to show for each $i_{N_{NLI}} \in dom(\omega_\tau)$ that $\tau(i_{N_{NLI}}(C_{N_{NLI}})) = \omega_\tau(i_{N_{NLI}})(C_{NatLog})$. The point here is that the two causal processes unfold in the same way, under any intervention. Indeed, pick any $i_{N_{NLI}}$ and suppose that $i_{NatLog} = \omega_\tau(i_{N_{NLI}})$. We know that $i_{NatLog}$ fixes values for the variables in $\mathbf{V}^{input}$, and likewise that $i_{N_{NLI}}$ fixes values for the variables $R_{11}, \ldots, R_{1m}$. Any other variables fixed by $i_{NatLog}$ from among $N, \mathrm{QP}_{Subj}$ will likewise correspond (via the component functions of $\tau$) to values of $L$ and $O$. We merely need to observe that any variables that are *not* set by $i_{NatLog}$ and $i_{N_{NLI}}$ will still correspond via the appropriate $\tau$-component, given their settings in $i_{NatLog}(C_{NatLog})$ and $i_{N_{NLI}}(C_{N_{NLI}})$. The intervention experiments on $N_{NLI}$ that we are assuming were successful were devised precisely to guarantee this.

We have thus fulfilled the three requirements and shown that $C_{NatLog}$ is an abstraction of $C_{N_{NLI}}$.