# OpenReview forum: "Causal Abstractions of Neural Networks"
_NeurIPS.cc/2021/Conference — NeurIPS 2021 Poster_

### Official Review · Reviewer_C38X · 2021-06-28

**Rating:** 6
**Confidence:** 4

**Summary:**

This paper proposes a method for identifying causal structures captured within neural networks. The proposed method does so by aligning a causal model with the neural networks’ prediction using interchange interventions. The authors demonstrate the efficacy of this method using the MQNLI corpus, where they show that a BERT-based model is compatible with a natural logic causal model such as the one originally used to generate the data.

Thus, the main contribution of the paper is a methodology for testing hypotheses regarding underlying causal structures learned in neural networks, which works well on an automatically generated natural language inference dataset.



**Limitations And Societal Impact:**

This paper can potentially have a positive societal impact, as understanding the causal structure engrained in neural networks can uncover hidden biases and fairness issues which govern the model’s behavior.

There are certainly limitations to this work, such as the reliance on a user’s ability to design a causal graph which might explain the neural network, and the inability of the method to provide causal identification besides fitting some potential model to the data. However, the paper provides a way forward for potentially explaining where causal knowledge is captured in neural networks.


**Main Review:**

This paper presents a complete process for generating hypotheses regarding possible causal structures governing the behavior of neural networks, and for testing their fit to the model's observed behavior. While it does require advanced prior task-related knowledge regarding possible causal structures which might be learned by the neural network, the proposed method is useful for tasks which could rely on such structure, such as many syntactic and semantic tasks in NLP.

The paper presents some originality in its attempt to provide causal abstractions and not only estimate causal effects, it is essentially a method for testing specific hypotheses regarding the interaction of model components. For researchers who are interested in estimating causal effects on neural networks and not in what specific model components (i.e. layers, neurons) are driving behavior, there are existing causal explanation methods that have already suggested possible solutions.

Still, this is an interesting paper that provides a clear and sound evaluation, and can surely contribute to researchers interested in uncovering where in the model a specific reasoning might occur (similarly to Causal Mediation Analysis). If rephrased as a contribution for reasoning about a specific model reasoning structure and not as a contribution for causal effect estimation, I think that this paper has a clear contribution to the literature. The current state of the paper, where it is presented as a tool for generating causal explanations (such as CausaLM), is somewhat misleading. Without comparing examples to counterfactuals, it is simply not true to state that this method truly uncovered causal effects.



**Time Spent Reviewing:**

4

---

> ### Author Response · Authors · 2021-08-08
> **Reviewer Response**
>
> Thank you for your thoughtful engagement with our work! We appreciate it greatly. We believe you hit the nail on the head by describing our method as a
>
> “complete process for generating hypotheses regarding possible causal structures governing the behavior of neural networks, and for testing their fit to the model's observed behavior”
>
> And we hope that our general response addressed your concern that there is a
>
> “reliance on a user’s ability to design a causal graph which might explain the neural network”
>
> ## Causal Abstraction Analysis Provides Causal Explanations by Evaluating Counterfactual Claims
>
> We would also like to address your concerns about whether our method is able to provide causal explanations and whether it evaluates counterfactual claims. We are confident that our method does both of these things!
>
> Specifically, we hope to convince you that our causal abstraction analysis method is indeed able to provide causal explanations of model behavior for the very same reason that a method like CausaLM provides causal explanations and that it does this by evaluating a number of counterfactual claims about the neural network.
>
> Quote: “The current state of the paper, where it is presented as a tool for generating causal explanations (such as CausaLM), is somewhat misleading. Without comparing examples to counterfactuals, it is simply not true to state that this method truly uncovered causal effects.”
>
> Fundamentally, interchange interventions evaluate counterfactual claims, answering the question of how a model would have behaved if a particular intervention were done. For this reason, our method is absolutely able to generate causal explanations, in the very same way that CausaLM generates explanations through interventions. Specifically, we generate a casual explanation by evaluating a number of counterfactual claims about neural network behavior by performing interventions on the network. However, CausaLM is limited to interventions on inputs, while we perform intermediate interventions as well. (as we note in our related work Line 76). Thus, in broad terms, we agree with your core tenets about what is needed for causal analysis in the relevant sense, and we are confident that our method adheres to them.
>
>
> ## Questions/Comments
>
> Quote: “If rephrased as a contribution for reasoning about a specific model reasoning structure and not as a contribution for causal effect estimation, I think that this paper has a clear contribution to the literature.”
>
>
> We did not intend to present our paper as a contribution for causal effect estimation, but instead wished to present it as a method for hypothesizing and testing the high level structure in neural models. In fact, we only used the term causal effect once in the paper, and that was to describe related work. In our next version, we will look for additional ways to make this clearer.

---

### Official Review · Reviewer_iJ4z · 2021-07-13

**Rating:** 6
**Confidence:** 4

**Summary:**

The paper presents a structural analysis tool grounded in the causal abstraction theory, through which it attempts to identify the causal properties governing the inference of the given task. This method is an alternative to probing, which involves training a supervised classifier to identify the causal relationships in many NLP tasks. Using this causal abstraction analysis tool, the authors study MQNLI corpus, which is a specifically constructed tree-structured corpus for natural language inference, grounded in natural logic. Through various experiments, the authors observe that BERT-based model approximately realizes the causal properties of the data generating process better than the baseline LSTM model.



**Limitations And Societal Impact:**

- The authors have not indicated any limitations of their study in the paper.
- In my opinion, the work should not result in any negative societal impact or ethical concerns. Thus, it does not require any further ethics review.



**Main Review:**

The paper presents novel, original method of identifying the causal mechanisms in play in neural networks for reasoning. This paper provides clear distinction of its methods from the prior work of probing literature and attribution methods using integrated gradients. The paper also provides good coverage of the related work which helps the reader to understand the necessary background.

The paper begins with a formal description of the hypothesis-alignment-intervention process. Certain parts of this section is not clear. For example, it is not clear (in L158 to L160) why z is not causally represented at L1 even if L1 depends on z in the MLP. The comparison with attribution based methods following the above argument is not clear, as a scalar value of 0 assigned with L1 with z would represent the same non-causal relationship the author is alluding to.

The paper is slightly hard to follow as it lacks clear examples. While the paper provides a good toy example describing the hypothesis formulation, alignment search and intervention process, those examples are not aligned with MQNLI pairs. In Figure 2, the causal structure is provided but it is not clear which pair it aligns with. Hence, following the arguments presented in Section 5.1 is hard. In Section 5.1, the interchange intervention possibly converts subsequence $e$ ("every happy person", "some happy baker") to ("every happy baker", "some \epsilon baker"). It is unclear how changing this subsequence renders the output label to neutral. A better example (including the entire sentence, not only the subsequence) would be helpful for the reader to understand the intervention process.

The analysis presented in the paper is also restricted to MQNLI - possibly due to the fact such interventional analysis would require practitioners to have access to the data generating process of the underlying dataset. This reduces the significance of the work a bit, although the sound theoretical motivation provided in this paper balances out the lack of downstream usability from an academic point of view. However, the paper only investigates two models - LSTM and a pre-trained BERT base - which reduces the significance of the results even academically. Out of these two models, LSTM model is much weaker compared to the BERT model - 46.32% compared to 88.50% - which makes the baseline of the causal abstraction argument much weaker anyway. The success of the high score of the BERT-based model is mostly due to pre-training, which the paper does not identify in the discussion. A better comparison would have been to isolate the non-pretrained models (including a BERT trained from scratch on MQNLI) and the pre-trained models (BERT and its family).

The alignment search phase raises a lot of questions as well. It is unclear in case of BERT where the representations are taken from. Due to multiple cascaded attentions in Transformer models, it is hard to pin down the representation of a word being the same location after several layers of self-attention. Thus, it is unclear how the hidden representations for tree nodes are extracted from BERT. Also, it is unclear how the word phrase representations (from nodes higher up in the tree) are used from BERT in the experiments.

The results and discussion section needs polish as well. While it is understandable that the causal intervention tool provides better signal than probing tasks, less discussion is provided on what exactly do we learn from the analysis of the alignment - i.e how important it is on the final prediction.

Overall, having lack of model exploration and restriction to MQNLI dataset does not exhibit condfidence that the proposed methodology can work in practice on other tasks. However, the theoretical experimental setup are still valuable for future practitioners to develop interventional analysis tools.

Edit: I have increased my score to 6 following the discussion with the authors, where they provide additional experiments and commit to clarify the examples and results.


**Time Spent Reviewing:**

8

---

> ### Author Response · Authors · 2021-08-08
> **Reviewer Response**
>
> Thank you  for your thoughtful feedback and engagement with our work! We appreciate it greatly. Your comments have resulted in immediate action on our part, fixing an example with an error in our text, and including additional baselines.
>
> ## Improved Example
> “The paper is slightly hard to follow as it lacks clear examples.” and “It is unclear how changing this subsequence renders the output label to neutra. A better example would be helpful.”
>
>
> Response: You are correct, a better example would be helpful, because we put the two MQNLI examples in the wrong order! In the main text, e and e’ should be swapped. Thanks so much for catching this! The example in the paper should be as follows
>
> C(e) = ("every happy baker", "no \epsilon baker")  = contradiction
>
> C(e’) =  ("every happy person", "some happy baker") = entailment
>
>
> C^{NP<-e’}(e) = C("every happy person", "no \epsilon baker")) = neutral
>
> We will update the text to fix this error!
>
> ## BERT w/ No Pretraining Baseline
> Quote: “A better comparison would have been to isolate the non-pretrained models (including a BERT trained from scratch on MQNLI”
>
>
> Response: We have now trained a non-pretrained BERT from scratch on MQNLI, and will include the results in the main results figure. We additionally evaluated a number of randomly generated alternative high-level hypotheses. See general response for details.
>
>
> ## Questions/Comments/Clarifications
> Quote: “For example, it is not clear (in L158 to L160) why z is not causally represented at L1 even if L1 depends on z in the MLP.”
>
>
> Response: W is causally represented at L_1 and W simply copies the value of Z, so Z is causally represented at L1. We hope this clears up any confusion, and we will clarify this in our main text!
>
> Quote: “In Figure 2, the causal structure is provided but it is not clear which pair it aligns with”
>
> Response: The causal structure in figure 2 is the causal structure of a natural logic algorithm that computes the relation between a premise and hypothesis sentence in MQNLI. All of the premise and hypothesis sentences have the same structure, and so this structure applies to all premise hypothesis pairs.
>
> Quote: “The alignment search phase raises a lot of questions as well. It is unclear in case of BERT where the representations are taken from. Due to multiple cascaded attentions in Transformer models, it is hard to pin down the representation of a word being the same location after several layers of self-attention. Thus, it is unclear how the hidden representations for tree nodes are extracted from BERT. Also, it is unclear how the word phrase representations (from nodes higher up in the tree) are used from BERT in the experiments.”
>
> Response:  See the general response for full details. For each high-level causal model, we perform a full exhaustive search through a space of possible alignments that are defined in section 5.1 under the bolded header “Alignment search”. The following prose will be added to line 232 in the main text:
>
> “The BERT model we use has 12 layers, and each alignment search considers aligning the intermediate high-level variable with dozens of possible locations in the grid of BERT representations. For each alignment considered, we performed a full causal abstraction analysis and we report the results from the best alignments in Figure 3, and summarize the results from all alignments in Figure 4 and Appendix D. ”

---

> > ### Comment · Reviewer_iJ4z · 2021-08-24
> > **Examples still not clear**
> >
> > Thank you for your reply! However, the example you provided is still not clear to me. To understand it further:
> >
> > - e contains both sentences, "every happy baker" and "no \epsilon baker"
> > - e' contains "every happy person", "some happy baker"
> > - If I take the analogy drawn in L127, C^{S_1 <- a'}(a) = x' + y' + z, then we have two noun phrases from e` : "happy person", "happy baker" and injecting it to e -> "every happy person", "no happy baker", which is different than the example you provided? Please clarify where my understanding is incorrect!
> >
> > Thanks for providing the responses for my other questions and running the BERT no-pretraining baseline. I still have some doubts regarding the overall efficacy of this approach for understanding model behavior for general NLI datasets, however the experimental setup is still valuable in its own scope. I would be happy to increase my score to 6 if the examples are clearly described and referenced in the results as well to make the intervention approach more legible to future readers.

---

> > > ### Author Response · Authors · 2021-08-24
> > > **Response had a typo**
> > >
> > > Thank you for your continued engagement with the reviewing process!
> > >
> > > Embarrassingly, there was a typo in our response to you. We have edited the response  so that the interchanged example reads:
> > >
> > > "Every happy person" "No happy baker"
> > >
> > > Which is what you, correctly, suggested it should be. Our apologies for the mistake, thanks for catching it! Do let us know if anything else is unclear.
> > >
> > > This exchange with you has certainly convinced us to take your advice and fully flesh out the details of this example in the main text, like we did with the toy math example. Hopefully this will increase the legibility and accessibility of our work. Thank you!

---

### Official Review · Reviewer_5z8f · 2021-07-15

**Rating:** 7
**Confidence:** 3

**Summary:**

1. The authors introduce a "causal abstraction" method for attributing high-level descriptions (combining representations and computations) to task-optimized neural networks.
2. They operationalize their method by performing causal abstraction analysis on a BERT model fine-tuned for MQNLI. They find substantial subsets of an MQNLI dataset for which their interchange intervention predictions are successful between any two pairs of examples.

**Limitations And Societal Impact:**

Yes.

**Main Review:**

The paper presents a very interesting interpretability method which goes beyond correlational and weaker causal analyses in the literature. However, I am concerned about a few practical issues that might affect the generalizability of the method. These are addressed in the "quality" section below.

I would be happy to see work like this published, and would recommend this paper for acceptance if the authors can adequately address these concerns. I may have misunderstood some parts of the analysis and would happily accept corrections as well.

## Quality

1. My main issue is with the lack of a comparison / baseline in the MQNLI evaluation. It's hard for me to know how to distinguish between a successful and unsuccessful analysis here. Concretely: at what level (and on what measure) would we confidently say that a proposed causal model is / is not a causal abstraction of the NN?My first thought on seeing the numbers in figure 3 was that this was not a success: for some syntactic constituents there are moderately-sized cliques of examples on which intervention is successful. But this doesn't hold more generally for the dataset, and the clique metric seems to degrade quickly as we move to the higher constituents covering larger spans of the input sentence.I would like the authors to address whether and how we can understand when a causal abstraction analysis has succeeded. (Are there reasonable a priori points on the evaluation metric scale which signal success? Are there statistical tests, e.g. a permutation test involving alternative forms of the causal model, which can give us a firm basis for judging success?)
2. A substantial practical issue with this method is inferring the alignment between variables in the causal model and "locations" in the neural model (I believe, following lines 89--90, this means subsets of neurons). As far as I can understand it, the authors did not actually run an alignment search in the MQNLI experiments, but rather stipulated that particular subsets of neurons should correspond to causal model variables (lines 218--231).I believe the generalizability of this method is severely limited by this alignment search method, as I understand it. There is very little information in the main text suggesting how this was carried out in their experiments, or gestures at how this would transfer to the general case. I enumerate my specific concerns below.I would like the authors to address these questions in the paper, and ideally demonstrate through their experiments that a full alignment search process can be tractable in a large-scale model.
  1. Such an alignment search involving models with complex causal structures would be expensive in terms of computation and data. For any node in the causal model, we would need to search among all subsets of neurons for correlations between causal model variables and neural representations. (I'm assuming correlation would be the measure, though I don't believe this is stated in the paper.) This would naturally require an extra development dataset separate from the one used for interchange experiments.
  2. The success criterion for alignment search is also an important question not addressed in the paper. At what degree of correlation do we conclude that an NN location is related to a causal model variable?

## Originality

I believe this principled approach to causal intervention is a novel contribution to the literature. However, it might be interesting to explore the relationship between this general method and related specific causal attribution analyses in NLP with a more implicit causal model -- for example, \[1\] run intervention analyses on language models to test how plurality and syntactic depth are represented.

## Minor comments and questions

1. 222--224: Why are there "two descendant tokens" for constituents such as Adj, Adv, V? As I understand from the example figure, you only consider inputs with one token for these constituents.
2. Fig. 3: which layer of the BERT model is used for these experiments?
3. More generally, I don't understand the relationship between fig. 3 and the "interchange intervention" experiments shown in fig. 4\. Fig. 3 shows the largest subset of examples on which C is a causal abstraction of the BERT model -- is this not the same as the size of the largest clique in the example graph with edges denoting interchange intervention success (i.e. the values shown in fig. 4a)?
4. I like the idea of using graph metrics for broad quantitative evaluation of the metric. But why is clique size the right metric? This seems to set a very high standard. Why not also incorporate a more permissive metric such as average degree, or visualize the full distribution over node degrees?

## References

1. Lakretz, Y., Kruszewski, G., Desbordes, T., Hupkes, D., Dehaene, S., & Baroni, M. (2019). The emergence of number and syntax units in LSTM language models. arXiv:1903.07435 \[cs\].[http://arxiv.org/abs/1903.07435](http://arxiv.org/abs/1903.07435)

**Time Spent Reviewing:**

3

---

> ### Author Response · Authors · 2021-08-08
> **Reviewer Response**
>
> Thanks so much for the thoughtful feedback and engagement with our work! We appreciate it greatly. Your suggestion to evaluate alternative forms of the high level causal model prompted us to run new experiments that will be included in our results as a further baseline! See the general response for details about that and your questions about the max clique size metric.
>
> We believe that the remaining concerns you trace to specific places where our own descriptions should have been clearer, leading to misunderstanding. In our general response, we have mapped out our strategy for addressing this weakness, including draft prose for the crucial revisions. Crucially, we do perform a search over a set of defined possible alignments, and the measure we evaluate alignments by is the interchange intervention clique size. In terms of experiment protocols, this seems comparable to what one does for other structural evaluation methods like probing and feature attribution.
>
>
> ## Alignment Search Success Criterion
>
> Quote: “The success criterion for alignment search is also an important question not addressed in the paper. At what degree of correlation do we conclude that an NN location is related to a causal model variable?”
> Response: The success criterion that we use for the alignment search is the interchange intervention success. We do our full causal abstraction analysis on every alignment we consider in our search, reporting on the alignments with the largest cliques. Correlation is not involved in this process.
>
> ## Minor comments and questions
>
> Quote: “222--224: Why are there "two descendant tokens" for constituents such as Adj, Adv, V? As I understand from the example figure, you only consider inputs with one token for these constituents”
> Response: The tree depicted in Figure 2 is an alignment between an MQNLI premise and hypothesis, where the intermediate nodes Adj, Adv, and V each have two leaf node token descendants, specifically the Adj/Adv/V in the premise and the Adj/Adv/V in the hypothesis.
>
> Quote: “Fig. 3: which layer of the BERT model is used for these experiments?”
> Response:We ran a full exhaustive search over alignments to each possible layer, and the best alignment is reported, where “best” is measured by interchange intervention success. This can be seen as comparable to layer-wise attribution or probing.
>
> Quote:“More generally, I don't understand the relationship between fig. 3 and the "interchange intervention" experiments shown in fig. 4. Fig. 3 shows the largest subset of examples on which C is a causal abstraction of the BERT model -- is this not the same as the size of the largest clique in the example graph with edges denoting interchange intervention success (i.e. the values shown in fig. 4a)?”
> Response: Figure 3 aggregates the results of our alignment search for each high level variable. Figure 4 reports the results for the alignment search of a single high level variable, namely NP_{Obj}, so each heatmap cell corresponds to a different alignment. We will use some of the extra space we have in our next version to clarify these connections.

---

> > ### Comment · Reviewer_5z8f · 2021-08-31
> > **Response response**
> >
> > Hi authors, thanks for your thorough responses, including the new control experiments! I'm excited to read more details about those experiments, and I hope they'll help to enrich the discussion started by this project.
> >
> > In light of these detailed extra controls, and resolving the good amount of misunderstanding that drove my original review, I'll adjust my score from 5 to 7.

---

### Official Review · Reviewer_Wd32 · 2021-07-18

**Rating:** 7
**Confidence:** 4

**Summary:**

see main

**Limitations And Societal Impact:**

see main

**Main Review:**

This paper aims to formalize and generalize the "interchange" approach to
interpreting computation in deep networks first proposed by Geiger et al.
(BlackBoxNLP 2020). In this experimental paradigm, an abstract causal model f of
the computation performed by a deep network model g is first hypothesized. Next,
alignments between individual variables in this causal model and components of
the deep network are guessed. Finally, the causal model + alignment combination
is validated by testing whether (in roughly Pearlian terms) f(y | x, do(z_f')) =
g(y | x, do(z_g')) for model outputs y and aligned latent / representation
pairs (z_f', z_g') extracted the application of f and g to alternative inputs x'.

The paper has three contributions:

(1) formalizing this analyis procedure in the language of constructive causal
    abstractions (Beckers & Halpern 2019).

(2) extending it to include a (brute-force) search over alignments rather than
    requiring researchers to fix these up-front

(2) applying it to the MQNLI dataset rather than the MONLI dataset originally
    used by Geiger et al '20

STRENGTHS

- Nice contextualization / formalization of the experimental paradigm of Geiger
  in terms of existing work on causal inference

- Useful example constructions for clarifying the expressive power of the
  probing paradigm in NLP

- Convincing evidence that the MacCartney-style inference is actually
  implemented (in a causal sense) by BERT models trained on MQNLI.

WEAKNESSES

- Approach itself is not new, and empirical findings largely overlap with the
  earlier Geiger paper.

- Most of the interesting content is in the appendix!

I'm quite conflicted about this paper. On one hand, drawing a link between
interchange interventions and the rest of the causal inference literature is
super useful. So are the example failure modes for probing and IG---while these
are simple constructions, I haven't seen them written down before and it would
be nice to have something to cite! But instead of focusing on this content, the
paper is mostly about analyzing the behavior of a single model trained on a
synthetic NLI dataset trained with an exotic form of supervision. I don't think
this finding is generalizable enough to be of interest to a general machine
learning audience. Meanwhile, the other methodological innovation on top of the
original Geiger paper is the addition of a search over candidate encodings,
which this paper largely sidesteps by manually pruning the search space by exploiting the regular structure of the
synthetic data. I think a stronger version of the paper would either (1) promote
appendices A and G to the main paper (and talk a little more about how we can
use other insights from causal inference to guide analysis of deep nets), (2) do
an empirical analysis of more than one model (e.g. a model trained on ordinary
NLI data), or (3) say a little bit more about how to automate the search (e.g.
by first building the matrix of correlations between causal variables and model
representations, and using these to prune the search space explored by
interchange interventions).

However, I'm absolutely willing to be persuaded that the paper is substantial enough to
accept in its current form, and would be interested in hearing more from the
authors about what they view as the delta (formally or empirically) relative to
Geiger '20 and the significance of the MQNLI results.

Some other thoughts:

SECTION 3

While the example in Fig 1 is useful, it's a little odd for a NeurIPS paper to
contain *only* a worked example, and leave the general framework that is being
exemplified out of the paper entirely. Building on the general comment above, is
it possible to interleave the discussion in section 3 with any concrete
discussion of constructive abstractions / interchange interventions in general?
To get space back, Fig 3 could be shrunk, and the nearly 2 pages of related work
spread across sections 2 and 5 could probably be condensed a bit.

Since interchange success and clique size tell different stories
about where causally relevant information is located (Fig 4), can we say
anything meaningful about the relationship between the two, or to formal
criteria for constituting a causal abstraction?

SECTION 5

Is max clique size really the right measurement here? I'd obviously rather have
one giant cluster and one medium sized cluster vs one giant cluster and 500 tiny
ones, and you could look at something like the entropy of the induced cluster or
the probability that any two nodes are connected by an edge.

What's the motivation for requiring cliques to have *at least one* impactful
edge as opposed to something stricter? (e.g. start with the graph in which
*every* edge corresponds to a label-changing substitutions in the true
causal model, then just measure the fraction of these edges where interchange
causes a corresponding label change in the neural net).

MISC

Extremely fussy: use \citet in expressions like "the dataset of [CITE]"


**Time Spent Reviewing:**

2.5

---

> ### Author Response · Authors · 2021-08-08
> **Reviewer Response**
>
> Thank you so much for your deep engagement with this material. We appreciate your comments and criticisms, and our work is  better for them!  We address your comments about (1) the delta between Geiger et al. (2020) and this paper, (2) the role of the max clique size measurement in the general response.
>
> ## Appendix A will be moved to the Main Text
>
> Quote:“promote appendices A and G to the main paper (and talk a little more about how we can use other insights from causal inference to guide analysis of deep nets)”
> Response: We appreciate the encouragement to highlight the contributions of these appendices. In the case of appendix A, we whole-heartedly agree and will move  this material to the main text, making use of the extra page we would receive if accepted and reducing the main figure as suggested.
>
>
>
> ## Additional Main Text Prose Informally Explaining Appendix G
>
> “While the example in Fig 1 is useful, it's a little odd for a NeurIPS paper to contain only a worked example, and leave the general framework that is being exemplified out of the paper entirely. Building on the general comment above, is it possible to interleave the discussion in section 3 with any concrete discussion of constructive abstractions / interchange interventions in general”
> As for appendix G, we are conflicted about the suggestion to bring the formal general framework into the main text. We believe that the formal material on causal abstraction would be very enlightening to a particular audience, but discouraging to a wider AI/ML audience that isn’t well-versed in formal causal models. It was our hope that by providing the formal material in the appendix and providing informal descriptions alongside a worked example in the main text, we would achieve both accessibility and depth. However, in light of your encouraging comments, we will include additional prose in section 3 to better integrate the content of appendix G into the main without including the formalisms:

---

> > ### Comment · Reviewer_Wd32 · 2021-08-25
> > **comment**
> >
> > Thank you for the extremely thorough author response! I've increased my score. I put 6 rather than 7 because I still think this would be a stronger paper if it included some kind of experiment on non-synthetic data (or at least a non-NLI task), but my other concerns about the relationship with Geiger et al have been addressed and I'd be happy to see this work at NeurIPS.

---

> > > ### Comment · Reviewer_Wd32 · 2021-08-31
> > > **comment 2**
> > >
> > > Actually, I've spent some time thinking about this and I would like to advocate more strongly for acceptance. I have increased my score to a 7. My original review undervalued the interestingness of the search procedure that has already been implemented, and the new experiments address some of my concerns regarding the delta from the BlackBoxNLP paper. I do hope the authors consider a followup with an additional task!

---

> > > > ### Author Response · Authors · 2021-08-31
> > > > **Grateful for the high quality reviews, and excited about follow up projects!**
> > > >
> > > > Thank you so much! We benefitted greatly from all the feedback, and we are grateful to have such invested and attentive reviewers. We are passionate about this method and are currently hard at work applying it to new domains such as navigation and mathematical reasoning!

---

### Author Response · Authors · 2021-08-08
**General Response**

We would like to begin by thanking our reviewers for their thoughtful engagement with our work! In full seriousness, these were truly high-quality reviews with insightful comments and criticisms that improved the quality of our work. Thank you so much!

## Appendix A will be moved to the Main Text

Due to the encouragement from reviews, we will move our analytical probing and IG examples to the main text.

## Python Library for Causal Abstraction Analysis

As part of this paper, we will release a library for causal abstraction analysis that we developed for this work. This package allows for you to define causal models, alignments between causal models, and allows you to perform efficient batched interchange interventions on neural networks.

This was not mentioned in the submitted draft, but we should have highlighted it, and will update the next draft to do so.

## New Experiment Completed: BERT with no Pretraining Baseline

In response to reviewer concerns about what the role of pretraining is, we trained a BERT transformer model from scratch and find that this model is unable to solve the MQNLI generalization task, achieving comparable performance to our baselines at 49%. The inclusion of this baseline demonstrates the crucial role of pretraining in BERT’s success on this difficult generalization task, which we will add to the result section prose in Section 4.

## New Experiment that Will Completed Soon: Evaluating Alternative High Level models

In response to reviewer concerns about how we evaluate whether our analysis was successful, we follow the suggestion to use “alternative forms of the causal model, which can give us a firm basis for judging success”.

Specifically, we have randomly generated high-level models with a single intermediate variable that composes together a random subset of input for some fixed size N. For instance, one of the random high-level models where N=2 composes together the object adjective in the premise and the hypothesis adverb. By evaluating alternative hypotheses encoded by other high-level model, we can provide evidence that the BERT model we are analyzing indeed abstracts the natural logic models we consider and that our analysis was “successful”.

We are in the process of completing these experiments, and will update our response when they are complete, which should be in the next day or two.

## Clarifying The Alignment Search Procedures

There were several misunderstandings in reviews about the alignment search we performed, which we took as an indication that we need to clarify the prose around this topic which we will in our updated draft. Here we describe the central changes that we plan to make to the text.

For each high-level causal model, we perform a full exhaustive search through a space of possible alignments that are defined in section 5.1 under the bolded header “Alignment search”.  The following prose will be added to line 232 in the main text:

“The BERT model we use has 12 layers, and each alignment search considers aligning the intermediate high-level variable with dozens of possible locations in the grid of BERT representations. For each alignment considered, we performed a full causal abstraction analysis and we report the results from the best alignments in Figure 3, and summarize the results from all alignments in Figure 4 and Appendix D. ”

We would also like to point out that the need for an alignment is not unique to our analysis method. When conducting probing and IG experiments, you always have to choose a location to test, and typically a full grid of locations is evaluated.

## Difference relative to Geiger et al.  (2020):

1. We are releasing a python package allowing researchers to define their own high-level causal models, load in their neural networks, and perform interchange interventions. Geiger et al. (2020) provides no such generalized software.

2.  Interchange interventions were introduced at the end of Geiger et al. (2020) as an informal method with no theoretical grounding that intuitively tests the hypothesis that a computation described by a bit of pseudo code is being carried out by a neural network. The current paper integrates this methodological tool into a formal framework of causality and abstraction, which allows for a general application of this methodology in many settings.

3. This paper has an analytical comparison (Appendix A, which will be moved to the main text) and empirical comparison (the main results) between IG, probing, and causal abstraction analysis.

4. The MoNLI dataset used by Geiger et al. (2020) is a simple dataset that is easily learned by a biLSTM baseline (93% accuracy). All of the examples in MoNLI have a premise and hypothesis that are identical except for a single word. This adversarial dataset is simply not intended to be a difficult, interesting task, but instead a tool for analyzing and improving NLI networks.
By contrast, the MQNLI dataset used in this paper is a dataset generated by a highly complex natural logic causal model, with a generalization task that was unsolved until this paper. Only a BERT with pretraining is able to solve the task (See our updated Baseline). There is an explanandum here: Why can the BERT model solve this difficult generalization task? Our answer is that it partially implements a natural logic algorithm.

4. This paper implements a full grid search over a set of possible alignments, akin to what would be done for probing or feature attribution to obtain a full picture, while Geiger et al. (2020) hand choose a single alignment.

## The Metric of Clique Size

If all of our interchange interventions are successful, then our analysis is a definite success. However, partial success requires thoughtful interpretation, so we choose the metric of maximum clique size because measures involving cliques are the most theoretically well grounded in the causal abstraction literature. When we have complete success on a clique of input examples, we know that causal abstraction relation holds between the neural network and high-level model in full force on that subset of inputs. Furthermore, we require that a clique has at least one impactful edge, which removes all cases where the abstraction relation holds on a subset of inputs, but the high-level variable is not even used on that subset.

We will add the following prose in the main text:

“We choose to measure the largest clique with at least one impactful edge, because (1) the causal abstraction relation holds with full force on that clique, but other measures lack this theoretical grounding and (2) if a clique has at least one impactful edge, that guarantees the high-level variable is being used. ”

The reviewers are correct that other graph measures or evaluation metrics could also be useful. We believe that as long as a reasonable metric is chosen, we can evaluate success by comparing our models of interest to baselines.  Evaluation metrics for partial success in this methodology are an active topic of discussion for us, so we appreciate the suggestions!

## The Need for a High-Level Model that Explains Network Behavior

Some reviewers expressed concerns that this method might be restricted by the need to have a high-level causal model that explains network behavior. We would like to first note that the need for a hypothesis is not unique to our analysis method. For example, before conducting a probing experiment, you have to have a piece of information that you hypothesize is encoded by the network. We also would like to argue that having a high-level model for intelligent behaviors that AI models are acquiring is not uncommon. In fact, many such models can be naturally adapted from theoretical and empirical modeling work in linguistics and cognitive sciences. We are currently conducting separate research using this method to understand how neural networks perform navigational tasks, syntactic parsing, and arithmetic reasoning. The approach we present is quite general!

---

> ### Comment · Reviewer_5z8f · 2021-08-25
> **Update on baseline / alternative-high-level-model evaluations?**
>
> Hi authors, I'm curious to hear about the extra evaluations you mentioned ("New Experiment that Will Completed Soon" above). Are these tests complete? Could you please share a summary of the results?

---

> > ### Author Response · Authors · 2021-08-26
> > **New experiment is still coming, but took longer than expected**
> >
> > Thank you for you continued engagement in the reviewing process. We appreciate it!
> >
> > We initially evaluated a small number alternative high-level models finding some interesting results, however we concluded that a more systematic exploration of this baseline would be fruitful and illuminating. As such, we have been running additional experiments with alternative high-level models. The final batch of experiments are finishing up today and tomorrow, so we plan to provide a full summary of our results then. We appreciate your patience, these experiments take a while to run!
> >
> > Thanks again for suggesting this baseline, we think it is going to provide greater clarity to our results!!

---

> > ### Author Response · Authors · 2021-08-27
> > **Alternative high level model evaluations are now complete**
> >
> > The reviewers raised some very interesting questions about what constitutes a success in our setting, and related to this, what we should use in the way of baselines. In the original submission we proposed clique size as a measure of success, and our baselines were other models that did not succeed at our main task. We were then able to make a relative claim, showing that clique sizes for the successful BERT model were systematically larger than those for the baseline LSTM model.
> >
> > However, the reviewer suggests another very helpful angle on the issue. The suggestion is to compare not just across alternative models, but across alternative causal hypotheses. We have pursued this fruitful suggestion and report here on some new and informative results that have come from it.
> >
> > Recall that our most successful high-level model,  $C_{NP_{Obj}}$,  composes the leaves $(A^P_{Obj}, N^P_{Obj}, A^H_{Obj}, N^H_{Obj} )$ and uses this composition to predict the final output. For this model we saw a clique size of 383. Following the reviewer’s suggestion, we were interested in comparing this against other conceivable causal models. As a first diagnostic case, we thus explored what would happen if we just considered causal models “in the neighborhood” of $C_{NP_{Obj}}$, specifically those obtained by adding one leaf, or by removing one or two leaves in the composition.
> >
> > The results were as follows:
> >
> > | Alternative High level model |       clique size |
> > | :---        |   ---: |
> > | $A^P_{Obj}, N^P_{Obj}, A^H_{Obj}, N^H_{Obj} $ | $\mathbf{383}$ |
> > | $A^P_{Obj}, N^P_{Obj}, A^H_{Obj} $ | 319|
> > | $A^P_{Obj}, N^P_{Obj}, N^H_{Obj} $ | 157|
> > | $A^P_{Obj},A^H_{Obj}, N^H_{Obj} $ | 338|
> > | $N^P_{Obj}, A^H_{Obj}, N^H_{Obj} $ | 158|
> > | $A^P_{Obj}, N^P_{Obj} $ | 319 |
> > | $A^P_{Obj},  A^H_{Obj} $ | 141|
> > | $ N^P_{Obj}, A^H_{Obj}$ | 322 |
> > | $A^P_{Obj},  N^H_{Obj} $ | 316|
> > | $ N^P_{Obj}, N^H_{Obj} $ | 88|
> > | $A^H_{Obj}, N^H_{Obj} $ | 321|
> >
> > | Alternative High level model |       clique size |
> > | :---        |   ---: |
> > | $NP_{Obj} $ | $\mathbf{383}$ |
> > | $NP_{Obj} + Adj^P_{Subj}$ | 305 |
> > | $NP_{Obj} + N^P_{Subj}$ | 372|
> > | $NP_{Obj} + Neg^P$ | 149|
> > | $NP_{Obj} + Adv^P$ | 269|
> > | $NP_{Obj} + V^P$ | 356|
> > | $NP_{Obj} + Q^H_{Obj}$ | 162 |
> > | $NP_{Obj} + Adj^H_{Subj}$ | 134 |
> > | $NP_{Obj} + N^H_{Subj}$ | 120 |
> > | $NP_{Obj} + Neg^H$ |  344|
> > | $NP_{Obj} + Adv^H$ | 162|
> > | $NP_{Obj} + V^H$ | 134|
> > | $NP_{Obj} + Q^H_{Obj}$ | 120 |
> >
> >
> > While some of the most similar high-level models are not too far off, remarkably all of them result in smaller clique size, for many significantly so. Indeed, most of these alternative causal models have a clique size far less than half.
> >
> > There are no doubt other high-level models one could explore here, and we are excited about pursuing this additional angle further in future work. In any case, the present experiments give us further confidence that the clique sizes we see in our original experiments are indeed indicative. We would like to thank the reviewer again for prompting us to explore this!

---

### Decision · Program_Chairs · 2021-09-27

**Decision:**

Accept (Poster)

**Comment:**

I don't think this paper requires a long metareview. They introduce a method for aligning the predictions of a network with an induced causal model of the task's "dynamics". They show it works on some NLI tasks which is respectable (although I would have liked to see some other domain too... maybe something to do in future work!). The reviewers were excited about the work, and several were ready to champion its acceptance after discussion with the authors. I recommend acceptance, and hope that ACL's loss here will be NeurIPS' gain.